# RELIABILITY-ADJUSTED PRIORITIZED EXPERIENCE REPLAY*

**Leonard S. Pleiss**
Technical University Munich
Munich, 80331
`leonard.pleiss@tum.de`

**Tobias Sutter**
University St. Gallen
St. Gallen, 9000
`tobias.sutter@unisg.ch`

**Maximilian Schiffer**
Technical University Munich
Munich, 80331
`schiffer@tum.de`

## ABSTRACT

Experience replay enables data-efficient learning from past experiences in online reinforcement learning agents. Traditionally, experiences were sampled uniformly from a replay buffer, regardless of differences in experience-specific learning potential. In an effort to sample more efficiently, researchers introduced Prioritized Experience Replay (PER). In this paper, we propose an extension to PER by introducing a novel measure of temporal difference error reliability. We theoretically show that the resulting transition selection algorithm, Reliability-adjusted Prioritized Experience Replay (ReaPER), enables more efficient learning than PER. We further present empirical results showing that ReaPER outperforms both uniform experience replay and PER across a diverse set of traditional environments including several classic control environments and the Atari-10 benchmark, which approximates the median score across the Atari-57 benchmark within one percent of variance.

## 1 INTRODUCTION

Reinforcement Learning (RL) agents improve by learning from past interactions with their environment. A common strategy to stabilize learning and improve sample efficiency is to store these interactions – called transitions – in a replay buffer and reuse them through experience replay to increase sample-efficiency. When using experience replay, the agent obtains mini-batches for training by sampling transitions from the replay buffer. Mini-batches are traditionally obtained using random sampling. However, a prioritization schemes can help to select more informative transitions, improving convergence speed and, ultimately, agent performance significantly (Schaul et al. (2015)). As such, the sampling scheme constitutes a performance-crititcal component for modern reinforcement learning agents leveraging experience replay (Hessel et al., 2017).

Among proposed prioritized sampling schemes, PER remains the most widely used (see Appendix C). PER was introduced in Schaul et al. (2015). It samples transitions in proportion to their absolute Temporal Difference Error (TDE), which measures the distance between predicted and target Q-values. Accordingly, PER follows the rationale that transitions with higher absolute TDEs bear higher learning potential. While this rationale is intuitive, the TDE is a biased proxy as both the predicted and the target Q-value are approximations. Hence, prioritizing transition selection based on absolute TDEs can misdirect learning, potentially leading to degrading value estimates, if the target Q-value is itself inaccurate (Panahi et al., 2024; Carrasco-Davis et al., 2025). Such inaccurate targets may dampen convergence or in the worst case deteriorate final policy performance.

To address the bias while retaining the efficient transition selection, we propose ReaPER, an enhanced experience replay strategy that extends PER by weighting the TDE with a measure of target Q-value reliability. This design is motivated by the observation that, when the agent's estimation of future states is inaccurate, the corresponding target Q-values become unreliable. In such cases, the TDE

ceases to be a dependable indicator of a transition's learning potential, leading to ineffective or even detrimental updates. By explicitly accounting for reliability, ReaPER preserves the advantages of PER over uniform experience replay while mitigating the negative impact of misleading priorities, resulting in consistent performance improvements.

**Intuition** The rationale behind our reliability estimate becomes particularly intuitive in game environments such as Go, Chess, or Tic Tac Toe. Consider a player assessing the current board position: if they lack a reliable understanding of how the game might unfold, their evaluation of the current state's value is likely inaccurate. As shown in Figure 1, states closer to terminal outcomes (i.e., near the end of the game) involve fewer remaining moves, making it easier for the agent to

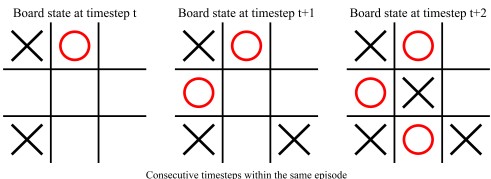

Figure 1: Subsequent states from a Tic Tac Toe game from the perspective of the agent placing circles. Board state $t + 2$ is terminal. States $t$ and $t + 1$ are losing under optimal play. For an inexperienced player, recognizing that $t + 1$ is a losing state is generally easier than recognizing $t$ as such. However, once $t + 1$ is understood as losing, identifying $t$ as lost becomes more straightforward. In general, accurately assessing $t + 1$ is a prerequisite for reliably assessing $t$—especially when learning the game without explicit knowledge of rules or win conditions. As long as the agent's assessment of $t + 1$ is flawed, its evaluation of $t$ remains unreliable.

accurately estimate their values. Early-game states, in contrast, rely on longer and more uncertain rollouts. Thus, value estimates tend to be more reliable as one moves closer to the end of an episode. This observation implies a hierarchical dependency in the learning of transitions within a trajectory, wherein the accurate estimation of earlier state-action values is conditioned on the agent's ability to infer and propagate information from later transitions. Consequently, we suggest that the reliability of target values – and by extension, of TDEs – should factor into experience replay prioritization.

**State of the Art** Experience replay has been an active field of research for decades. After its first conceptualization by Lin (1992), various extensions, analyses and refinements have been proposed (e.g., Andrychowicz et al. (2017); Zhang & Sutton (2017); Isele & Cosgun (2018); Rolnick et al. (2018); Rostami et al. (2019); Fedus et al. (2020); Lu et al. (2023)).

Central to our work is an active stream of research exploring optimized selection of experiences from the replay buffer. The most notable contribution in this stream so far was PER (Schaul et al., 2015; Panahi et al., 2024). In essence, PER proposes to use the absolute TDE as a sampling weight, which allows to select transitions with high learning potential more frequently compared to a uniform sampling strategy. Various papers built upon the idea of using transition information as a transition selection criterion: Ramicic & Bonarini (2017) explored an entropy-based selection criterion. Gao et al. (2021) proposed using experience rewards for sample prioritization. Brittain et al. (2019) introduced Prioritized Sequence Experience Replay, which extends PER by propagating absolute TDEs backwards throughout the episode before using them as a sampling criterion. Zha et al. (2019) and Oh et al. (2021) proposed dynamic, learning-based transition selection mechanisms. Yet, the proposed approaches have not replaced PER: PER remains the only prioritized sampling strategy that is widely adopted by state-of-the-art RL algorithms. For a comprehensive review of the relevant literature and a systematic breakdown of this claim, we refer to Appendix C.

**Contribution** We propose ReaPER, a novel experience replay sampling scheme that improves upon PER by reducing the influence of unreliable TD targets, ultimately leading to more stable learning and better policy performance. Specifically, our contribution is threefold: first, we propose the concept of target Q-value and TDE reliability and introduce a reliability score based on the absolute TDEs in subsequent states of the same trajectory. Second, we present formal results proving the effectiveness of the reliability-adjusted absolute TDE as a transition selection criterion. Third, we leverage the theoretical insights and the novel reliability score to propose ReaPER, a sampling scheme facilitating

more effective experience replay. The proposed method is algorithm-agnostic and can be used within any off-policy RL algorithm.

To substantiate our theoretical findings, we perform numerical experiments comparing ReaPER to PER across various traditional RL environments, namely CARTPOLE, ACROBOT, LUNARLANDER and the ATARI-10 benchmark, which recovers 99.2% of variance within the median score estimate of the full Atari-57 benchmark (Aitchison et al., 2022). We show that both prioritized sampling strategies outperform uniform experience replay, and further show that ReaPER consistently outperforms PER. Specifically, in environments of lower complexity like CARTPOLE, ACROBOT and LUNARLANDER, ReaPER reaches the maximum score between 16.6% and 32.6% faster than PER. In environments of higher complexity, exemplified by the ATARI-10 benchmark, ReaPER achieves a 22.97% higher median peak performance. In a partially observable variant of the ATARI-10 benchmark, the performance gap widens, with ReaPER achieving a 34.98% median improvement.

## 2 PROBLEM STATEMENT

We consider a standard Markov decision process (MDP) as usually studied in an RL setting(Sutton & Barto, 1998). We characterize this MDP as a tuple $(\mathcal{S}, \mathcal{A}, P, r, \gamma, p)$, where $\mathcal{S}$ is a finite state space, $\mathcal{A}$ is a finite action space, $P : \mathcal{S} \times \mathcal{A} \to \Delta(\mathcal{S})$ is a stochastic kernel, $r : \mathcal{S} \times \mathcal{A} \to \mathbb{R}$ is a reward function, $\gamma \in (0, 1)$ is a discount factor, and $p \in \Delta(\mathcal{S})$ denotes a probability mass function denoting the distribution of the initial state, $S_1 \sim p$. At time step $t$, the system is in state $S_t = s \in \mathcal{S}$. We denote by $S_t$ and $A_t$ the random variables representing the state and action at time $t$, and by $s \in \mathcal{S}$ and $a \in \mathcal{A}$ their respective realizations. If an agent takes action $A_t = a \in \mathcal{A}$, it receives a corresponding reward $r(s, a)$, and the system transitions to the next state $S_{t+1} \sim P(\cdot|s, a)$. We define the random reward at time $t$ as $R_t = r(S_t, A_t)$. The agent selects actions based on a policy $\pi : \mathcal{S} \to \mathcal{A}$ via $A_t = \pi(S_t)$.

Let $\mathbb{P}_p^\pi(\cdot) = \text{Prob}(\cdot \mid \pi, S_1 \sim p)$ denote the probability of an event when following a policy $\pi$, starting from an initial state $S_1 \sim p$, and let $\mathbb{E}_p^\pi[\cdot]$ denote the corresponding expectation operator. We consider problems with finite episodes, where $n$ expresses the number of transitions within the episode. Let $G_t$ denote the discounted return at time $t$, $G_t = \sum_{i=t}^n \gamma^{i-t} R_i$. We define the Q-function (or action-value function) for a policy $\pi$ as

$$Q^\pi(s, a) = \mathbb{E}_p^\pi \left[ G_t \mid S_t = s, A_t = a \right] = \mathbb{E}_p^\pi \left[ \sum_{i=t}^n \gamma^{i-t} R_i \mid S_t = s, A_t = a \right]. \tag{1}$$

The ultimate goal of RL is to learn a policy that maximizes the Q-function, leading to $Q^\star(s, a) = \max_\pi Q^\pi(s, a)$. The policy is gradually improved by repeatedly interacting with the environment and learning from previously experienced transitions. A transition $C_t$ is a 5-tuple, $C_t = (S_t, A_t, R_t, S_{t+1}, d_t)$, where $d_t$ is a binary episode termination indicator, $d_t = \mathbb{1}_{t=n}$. One popular approach to learn $Q^\star$ is via Watkins' Q-learning (Watkins, 1989; Watkins & Dayan, 1992), where Q-values are gradually updated via

$$Q(S_t, A_t) \leftarrow Q(S_t, A_t) + \eta \cdot \delta_t \tag{2}$$

in which $\eta \in (0, 1]$ is the learning rate and $\delta_t$ the TDE, $\delta_t = Q_{\text{target}}(S_t) - Q(S_t, A_t)$ with $Q_{\text{target}}(S_t) = R_{t+1} + (1 - d_{t+1}) \cdot \gamma \cdot \max_a Q(S_{t+1}, a)$. For brevity of notation, we refer to the absolute TDE as $\delta_t^+ = |\delta_t|$.

In practical RL deployments, when the Q-function is approximated with a neural network, this framework has been augmented with several influential extensions, including Double Deep Q-Network (DDQN), target networks, and experience replay (see Appendix B). Experience replay is commonly employed to stabilize and accelerate learning. Transitions collected through agent-environment interaction are stored in a finite buffer $\mathcal{H} = \{C_t\}_{t=1}^N$, from which mini-batches $\mathcal{X} \subset \mathcal{H}$ of fixed size $|\mathcal{X}| = k$ are sampled to update the Q-function. The sampling distribution over the buffer, denoted by $\mu \in \Delta(\mathcal{H})$, determines the likelihood $\mu(C_t)$ of selecting transition $C_t \in \mathcal{H}$ when constructing $\mathcal{X}$. In uniform experience replay, $\mu$ is the uniform distribution, whereas in PER (Schaul

et al., 2015), transitions are sampled according to scalar priority values, derived from the absolute TDE $\delta_t^+$.

Empirical evidence suggests that the effectiveness of the learning process is sensitive to the choice of $\mu$, i.e., sampling transitions with high learning potential can improve convergence speed and final performance. However, designing an optimal or near-optimal sampling distribution remains an open problem. With this work, we aim to contribute to closing this gap by defining and efficiently approximating a sampling distribution $\mu^\star$ that maximizes learning progress using experience replay.

## 3 METHODOLOGY

In the following, we provide the methodological foundation for ReaPER. We first introduce a reliability score for absolute TDEs, which we use to derive a TDE-based reliability-adjusted transition sampling method. We then provide theoretical evidence for its efficacy.

### 3.1 RELIABILITY SCORE

In bootstrapped value estimation, as in Q-learning, the target value

$$Q_{\text{target}}(S_t) = R_{t+1} + \gamma \cdot (1 - d_{t+1}) \cdot \max_a Q(S_{t+1}, a) \tag{3}$$

relies on the current estimate of future values. Consequently, the quality of an update to $Q(S_t, A_t)$ depends not only on the magnitude of the absolute TDE $\delta_t^+$, but also on the reliability of the target value $Q_{\text{target}}(S_t)$.

We define the reliability of a target Q-value as a measure of how well it approximates the true future return from a given state-action pair. Intuitively, a target value is reliable if it decreases the distance between $Q^\star(S_t, A_t)$ and $Q(S_t, A_t)$. Conversely, a target is unreliable if training on it increases the distance between $Q^\star(S_t, A_t)$ and $Q(S_t, A_t)$.

To motivate this concept and formalize its operational consequences, we consider a single episode consisting of transitions $(C_1, \ldots, C_n)$, from initial state $S_1$ to terminal state $S_{n+1}$. We highlight three key observations that explain how reliability varies along the trajectory and how it can be used to improve sampling:

**Observation 3.1** (Unreliable targets can degrade learning). *$Q_{target}(S_t)$ depends on the estimate $Q(S_{t+1}, \cdot)$ for $t \in \{1, \ldots, n-1\}$, which may be inaccurate. If an update is based on a poor target value, the resulting $Q(S_t, A_t)$ may diverge from $Q^\star(S_t, A_t)$, thereby degrading the estimate.*

**Observation 3.2** (Terminal transitions induce reliable updates). *For terminal transitions, the target is given directly by the environment, i.e., $Q_{target}(S_n) = R_n$. This target is exact, implying that the corresponding TDE accurately reflects the deviation from the ground truth Q-value. Thus, updates based on terminal transitions are guaranteed to shift $Q(S_n, A_n)$ towards $Q^\star(S_n, A_n)$ if $\delta_n^+ > 0$.*

**Observation 3.3** (Reliability propagates backward). *An accurate update to $Q(S_t, A_t)$ improves the accuracy of $Q_{target}(S_{t-1})$ and earlier targets, which recursively depend on it. Therefore, updating transitions near the end of the episode helps improving the reliability of $Q_{target}$ for earlier transitions.*

These observations highlight a temporal hierarchy in transition learning: *Learning later transitions before learning earlier transitions appears advantageous.* On the one hand, later targets rely on fewer estimated quantities and are therefore more reliable. On the other hand, learning later transitions positively impacts the target reliability for earlier transitions. Furthermore, a high TDE indicates a misunderstanding of game dynamics for a given transition, thus rendering the value estimation in predecessor transitions – which rely on the understanding of the value dynamics in the subsequent rollout – less reliable. We therefore aim to resolve TDEs back-to-front. This motivates defining the reliability of $Q_{\text{target}}(S_t)$ inversely related to the sum of future absolute TDEs,

$$\mathcal{R}_t = 1 - \frac{\sum_{i=t+1}^n \delta_i^+}{\sum_{i=1}^n \delta_i^+}. \tag{4}$$

Using this definition, we propose the *reliability-adjusted TDE*

$$\Psi_t = \mathcal{R}_t \cdot \delta_t^+, \tag{5}$$

as a sampling criterion for selecting transitions during experience replay. High values of $\Psi_t$ correspond to transitions that promise large updates and have reliable target values. Sampling weights $p$ can be obtained by normalizing the sampling criterion with the sum of $\Psi$ over all transitions.

## 3.2 FORMAL ANALYSIS

We consider a set of transitions that constitutes a single complete trajectory of length $n$, $\mathcal{D} = \{C_t\}_{t=1}^n$, where $C_t = (S_t, A_t, R_t, S_{t+1}, d_t)$.

Updates are based on the TDE $\delta_t = Q(S_t, A_t) - Q_{\text{target}}(S_t)$, using the standard bootstrapped target

$$Q_{\text{target}}(S_t) = R_t + \gamma(1 - d_t)\max_a Q(S_{t+1}, a). \tag{6}$$

**Convergence** A critical factor to ensure convergence in Q-learning is the alignment between the TDE and the true value estimation error $Q(S_t, A_t) - Q^\star(S_t, A_t)$. When the bootstrapped target is biased, meaning $Q_{\text{target}}(S_t) \neq Q^\star(S_t, A_t)$, the update direction may become misaligned, potentially worsening the value estimate.

We defer the formal misalignment analysis to Lemma D.1 and Lemma D.2 in Appendix D.1. In essence, the expected change in squared true value estimation error due to an update of the value function approximator under a sampling strategy $\mu$ can be decomposed into three components, the TDE variance, the true squared error, and the bias-error interaction

$$\mathbb{E}_\mu[\Delta|Q(S_t, A_t) - Q^\star(S_t, A_t)|^2] = \eta^2 \underbrace{\sum_{t=1}^n \mu_t \mathbb{E}[\delta_t^2]}_{\text{TDE variance}} - 2\eta \underbrace{\sum_{t=1}^n \mu_t \mathbb{E}[e_t^2]}_{\text{True squared error}} + 2\eta \underbrace{\sum_{t=1}^n \mu_t \mathbb{E}[e_t \varepsilon_t]}_{\text{Bias-error-interaction}}, \tag{7}$$

where $e_t$ denotes the true value error, and $\varepsilon_t$ denotes the target bias,

$$e_t = Q(S_t, A_t) - Q^\star(S_t, A_t), \quad \varepsilon_t = Q_{\text{target}}(S_t) - Q^\star(S_t, A_t). \tag{8}$$

By focusing on large TDEs, PER aims to sample transitions with higher true squared error more frequently, thus resolving errors faster and improving efficiency over uniform sampling. In the following, we show that our ReaPER sampling scheme additionally controls the target bias, thereby minimizing the bias-error interaction, while also preserving the advantages of PER. This allows ReaPER to increase sampling-efficiency over PER. To do so, we base the following formal analyses on a key assumption relating target bias to absolute downstream TDEs.

**Assumption 3.4** (Target Bias via Downstream TDEs). *Along an optimal trajectory, the target bias $\varepsilon_t$ for each transition $C_t$ satisfies*

$$|\varepsilon_t| \leq \lambda \sum_{i=t+1}^n \delta_i^+, \tag{9}$$

for some $\lambda \leq 1$, where $\delta_i^+ = |\delta_i|$. This assumption formalizes the intuition that bootstrapped targets primarily inherit bias from inaccuracies in future predictions. It reflects standard TD-learning dynamics under sufficient exploration and function approximation stability. While – similar to the assumptions made in standard convergence proofs for RL – this assumption may theoretically be violated during the early phases of training, it tends to hold in practice, especially once value estimates stabilize, as we show in Appendix E. In a nutshell, Assumption 3.4 captures the intuition that target bias predominantly arises from unresolved downstream TDEs, reflecting a local perspective on TD-learning dynamics. Unlike classical convergence proofs that require global exploration and decaying learning rates, our assumption focuses on bounding the bias along observed trajectories during finite-sample learning, making it more applicable to practical deep RL. We refer the interested reader to Appendix D.2 for a detailed discussion.

Under Assumption 3.4, the reliability score $\mathcal{R}_t$ –representing the fraction of downstream TDE within a given trajectory – bounds the normalized target bias. This is captured in the following lemma.

**Lemma 3.5** (Reliability Bounds Target Bias). *Under Assumption 3.4,*

$$|\varepsilon_t| \leq \lambda(1 - \mathcal{R}_t)\sum_{i=1}^n \delta_i^+. \tag{10}$$

For the proof of Lemma 3.5, we refer to Appendix D.3. Lemma 3.5 expresses that transitions with higher reliability scores exhibit lower target bias and thus yield more trustworthy TDEs. This finding motivates selecting training transitions not just by TDE magnitude, but by a combination of TDE magnitude and reliability — as implemented in ReaPER.

Building on the established relationship between reliability and target bias, we derive the following convergence hierarchy.

**Proposition 3.6** (Convergence Hierarchy of Sampling Strategies). *Under Assumption 3.4 and given a fixed learning rate $\eta$, ReaPER ($\mu_t \propto \mathcal{R}_t \delta_t^+$) yields lower expected Q-value error than standard PER ($\mu_t \propto \delta_t^+$), which in turn outperforms uniform sampling,*

$$\mathbb{E}[||Q_T^{(\text{Uniform})} - Q^\star||^2] \geq \mathbb{E}[||Q_T^{(\text{PER})} - Q^\star||^2] \geq \mathbb{E}[||Q_T^{(\text{ReaPER})} - Q^\star||^2], \qquad (11)$$

*where $\mathbb{E}$ denotes the expectation across training runs.*

The corresponding proof, detailed in Appendix D.4, formally compares the expected error decrease terms under different sampling distributions, using Lemma 3.5 to bound the bias-error-interaction. While we limit Proposition 3.6 to optimal policies for brevity of notation, we can straightforwardly extend it to suboptimal policies.

**Remark 3.7** (Extension to suboptimal policies). *If the agent follows a fixed but suboptimal policy, Assumption 3.4 can be relaxed to include an additive policy-induced bias term $\zeta \geq 0$, yielding*

$$|\varepsilon_t| \leq \lambda \sum_{i=t+1}^{n} \delta_i^+ + \zeta. \qquad (12)$$

*In this case, ReaPER still improves sampling efficiency in expectation, although the achievable Q-value accuracy is lower-bounded by the policy suboptimality $\zeta$. For further details, we refer the interested reader to Appendix D.5.*

Together, these results provide the theoretical foundation for ReaPER's design: By prioritizing transitions with high absolute TDE and high reliability, ReaPER selects relevant transitions while improving alignment with the true value error, leading to faster and more stable learning.

**Variance reduction** In the following, we show that the proposed sampling scheme, based on reliability-adjusted TDEs reduces the variance of the Q-function updates. As a first step towards this result, we analyze the theoretically optimal distribution to sample from in order to minimize the variance of the Q-function update step. Recall that Q-values are updated according to (2), where the TDE corresponding to a transition $C_t$ from the finite replay buffer $\mathcal{H}$ reads $\delta_t = Q_{\text{target}}(S_t) - Q(S_t, A_t)$.

We assume a fixed episode and treat the current $Q$-values as constants, focusing on analyzing the update variance induced by the sampling distribution $\mu$ over $\mathcal{H}$. The update variance can then be expressed as

$$\sum_{t=1}^{N} \mu_t \text{Var}[\delta_t] = \sum_{t=1}^{N} \mu_t \text{Var}[Q_{\text{target}}(S_t)] = \sum_{t=1}^{N} \mu_t \sigma_t^2, \qquad (13)$$

where the first equality follows from the definition of the TDE and the assumption that the current $Q$-values are constant. The second equality simply defines $\sigma_i^2 := \text{Var}[Q_{\text{target}}(S_i)]$ as the variance of the bootstrapped target for brevity.

**Proposition 3.8** (Variance reduction via reliability-aware sampling). *The distribution $\mu^\star$ minimizing the update variance (13) is given by*

$$\mu_t^\star \propto \frac{\delta_t^+}{\sigma_t^2} \text{ for all } t \in \{1, \dots, N\} \qquad (14)$$

For the proof of Proposition 3.8, we refer to Appendix D.6.

As a direct consequence of Proposition 3.8, we find that our proposed ReaPER sampling scheme is variance reducing if the reliability $\mathcal{R}$ is proportional to the inverse variance of the bootstrapped target.

As the true target $Q^\star$ remains constant, a significant proportion of variance across runs for a given state can be attributed to the target bias. As such, there exists a direct relationship between $\varepsilon$ and $\sigma^2$. Hence, under Assumption 3.4, it seems natural to assume $\mathcal{R} \propto \frac{1}{\sigma^2}$. Thus, ReaPER constitutes a reasonable proxy for the optimal inverse-variance weighted sampling strategy.

To provide further intuition for our formal results, we have conducted a supplementary simulation-driven analysis showing that ReaPER achieves optimal transition selection in a stylized setting, which we detail in Appendix F.

## 4 RELIABILITY-ADJUSTED PRIORITIZED EXPERIENCE REPLAY

We have thus far introduced the reliability-adjusted TDE and theoretically proven its effectiveness as a transition selection criterion. In the following, we propose ReaPER, the sampling algorithm built around the reliability-adjusted TDE. We give a distilled overview of the resulting sampling scheme in Algorithm 1. At its core, we create mini-batches by sampling from the buffer with $\Psi$ as the sampling weight. Specifically, at each training step $\tau \in \{1, \dots, T\}$, for every transition within the buffer, that is, $C_t$ for all $t \in \{1, \dots, N\}$, we update TDE reliabilities $\mathcal{R}_t$ and compute the transition selection criterion $\Psi_t$ (Algorithm 1, Line 5ff.). Based on $\Psi_t$, we sample $k$ transitions from the buffer $\mathcal{H}$ to create the next mini-batch $\mathcal{X}$ (Algorithm 1, Line 9ff.). For the full algorithm and an extended explanation, we refer to Appendix A.

---

**Algorithm 1:** Sampling transitions and updating the value function using ReaPER

**Input:** absolute TDEs $\delta^+$, episode vector $\phi$, current episode $\Phi$, batch size $k$, exponents $\alpha$, $\omega$ and $\beta$, replay buffer $\mathcal{H}$ of size $N$, policy weights $\theta$, maximum priority $p_{max} = 1$, budget $T$

1  **for** $\tau \in \{1, \dots, T\}$ **do**
2     Initialize accumulated weight change $\Delta = 0$ and empty batch $\mathcal{X} = \mathbf{0}^{(k)}$;
3     Add novel transitions to the buffer with maximum priority $p_{max}$ and set $\phi_t = \Phi$;
4     Compute maximum episodic sum of absolute TDEs, $F \leftarrow \max\limits_{t \in \{1, \dots, N\}} \left( \sum_{i=1}^{N} \delta_i^+ \cdot \mathbb{1}_{\phi_t = \phi_i} \right)$;
5     **for** $t \in \{1, \dots, N\}$ **do**          // Updating transition weights
6         Compute TDE reliabilities as in Formula 16;
7         Compute transition selection criterion $\Psi_t \leftarrow \mathcal{R}_t^\omega \cdot \delta_t^{+\alpha}$;
8         Compute transition priorities $p_t \leftarrow \frac{\Psi_t}{\sum_{i=1}^{N} \Psi_i}$;
9     **for** $m \in \{1, \dots, k\}$ **do**          // Sampling transitions
10         Sample a transition $C_j$ from $\mathcal{H}$ to add to batch $\mathcal{X}$ such that $\mathbb{P}(C_t = \mathcal{X}_m) = p_t$ for all $t \in \{1, \dots, N\}$;
11         Compute importance-sampling weight $w_j \leftarrow \frac{(N \cdot p_j)^{-\beta}}{\max_t w_t}$ for all $t \in \{1, \dots, N\}$;
12         Update $\delta_j^+$ and accumulate weight-change $\Delta \leftarrow \Delta + w_j \cdot \delta_j \cdot \nabla_\theta Q(S_j, A_j)$;
13     Update weights $\theta \leftarrow \theta + \eta \cdot \Delta$;
14     Update maximum priority $p_{max} = \max(p_{max}, \max(p))$;

---

Starting from the naive implementation of ReaPER, we require four technical refinements to obtain a functional and efficient sampling algorithm.

I. *Priority updates.* To consistently maintain an updated sampling weight $\Psi$, we track the TDEs and reliabilities of stored transitions throughout the training. As it is computationally intractable to re-calculate all TDEs on every model update, we implement a leaner update rule: As in PER, we assign transitions maximum priority when they are added to the buffer. Moreover, we assign the TDE of transition $C_t$ every time $C_t$ is used to update the Q-function. We assign the reliability of transition $C_t$ every time $C_t$ is used to update the Q-function, or if any other transition from the same episode is used to update the Q-function, as it leads to a change in the sum of TDEs and possibly the subsequent TDEs. We update the priority $\Psi$ when TDE or reliability are updated.

II. *Priority regularization.* As the TDE of a given transition may change by updating the model even without training on this transition, TDEs – and, in consequence, reliabilities – are not guaranteed to

be up-to-date. Thus, similar to Schaul et al. (2015), we introduce regularization exponents $\alpha \in (0, 1]$ and $\omega \in (0, 1]$ to dampen the impact of extremely high or low TDEs or reliabilities,

$$\Psi_t = \mathcal{R}_t^\omega \cdot \delta_t^{+\alpha}. \tag{15}$$

III. *Reliabilities for ongoing episodes.* As the sum of TDE throughout an episode is undefined as long as the episode is not terminated, so is the reliability. In these cases, we use the maximum sum of TDEs of any episode within the buffer to obtain a conservative reliability estimate.

For this, we introduce $\phi$, a vector of length $N$, where $\phi_t$ denotes the $t$-th position in $\phi$, which contains a scalar counter of the trajectory during which transition $C_t$ was observed. As such, $\phi$ functions as a positional encoding of transitions within the buffer. Specifically, it is used to identify all transitions that belong to the same trajectory. This positional encoding allows us to calculate conservative reliability estimates for a multi-episodic buffer.

We then define $\mathcal{R}_t$ as

$$\mathcal{R}_t = \begin{cases} 1 - \left( \frac{\sum_{i=t+1}^n \delta_i^+}{\sum_{i=1}^n \delta_i^+} \right) & \text{for transitions of terminated episodes} \\ 1 - \left( \frac{F - \sum_{i=1}^t \delta_i^+}{F} \right) & \text{for transitions of ongoing episodes} \end{cases} \tag{16}$$

where

$$F = \max_{t \in \{1, \dots, N\}} \left( \sum_{i=1}^N \delta_i^+ \cdot \mathbb{1}_{\phi_t = \phi_i} \right) \tag{17}$$

.

Importantly, this formulation of reliability focuses on within-episodic variance instead of inter-episodic comparability. It therefore deliberately does not account for differences in episode length, as this introduces bias in favor of shorter trajectories, which we found to be detrimental in practice.

VI) *Weighted importance sampling.* Finally, just as every other non-uniform sampling method, ReaPER violates the i.i.d. assumption. Thus, it introduces bias into the learning process, which can be harmful when used in conjunction with state-of-the-art RL algorithms. Similar to Schaul et al. (2015), we use weighted importance sampling (Mahmood et al., 2014) to mitigate this bias. When using importance sampling, each transition $C_t$ is assigned a weight $w_t$, such that

$$w_t = \left( \frac{1}{N} \cdot \frac{1}{p_t} \right)^\beta = \left( \frac{1}{N} \cdot \frac{\sum_{i=1}^N \Psi_i}{\Psi_t} \right)^\beta. \tag{18}$$

We use this weight to scale the loss and perform Q-learning updates using $\delta_t \cdot w_t$ instead of $\delta_t$.

## 5 NUMERICAL STUDY

We evaluated ReaPER against PER across a diverse set of continuous control and Atari environments. For continuous control, we considered the discrete action space environments from the Gymnasium library (Towers et al., 2024), namely CARTPOLE, ACROBOT and LUNARLANDER. For Atari, following prior work, we use ATARI-10 as a computationally efficient yet representative benchmark which recovers 99.2% of median score variance within the Atari-57 benchmark, ensuring relevance to broader Atari-57 evaluations without incurring prohibitive computational overhead (Aitchison et al., 2022). Across conditions, we used the same DDQN agent, neural architecture, and model hyperparameters for all experiments. We controlled for all sources of randomness using fixed seeds and compared algorithms using identical seeds per trial. Thus, the only variation between conditions stemmed from the experience replay algorithm. Full experimental details and hyperparameters are provided in Appendix G. For a detailed introduction to additional deep Q-learning–related concepts, we refer to Appendix B.

**Continuous control**    For each continuous control environment, we compared the performance between PER and ReaPER across 20 training runs. Training ended preemptively when a pre-defined score threshold was met (Towers et al., 2024).

Across all three environments, ReaPER consistently reached performance thresholds in fewer steps than both uniform replay and PER. In ACROBOT, this corresponded to improvements of $25.0\%$ and $16.6\%$, respectively. For CARTPOLE, ReaPER reduced the steps needed by $41.4\%$ and $32.6\%$, and a similar pattern held in LUNARLANDER, with gains of $37.1\%$ and $21.1\%$.

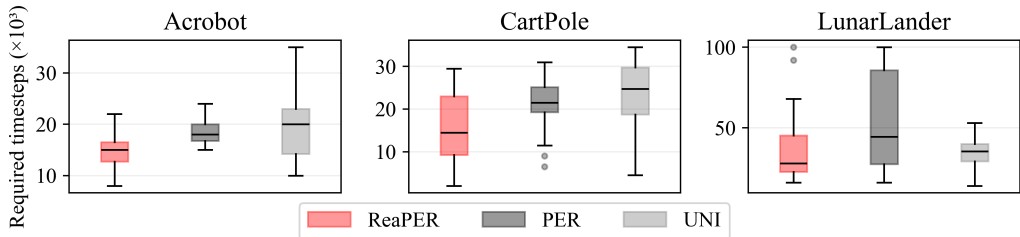

Figure 2: Proportion of training steps required by PER, uniform experience replay (UNI) and ReaPER to reach a pre-defined score thresholds given in Towers et al. (2024) across 20 runs in three traditional RL environments. The shaded region corresponds to the interquartile range (IQR), whiskers extend to $Q1 - 1.5 \cdot IQR$ and $Q3 + 1.5 \cdot IQR$, and the horizontal bar indicates the median score.

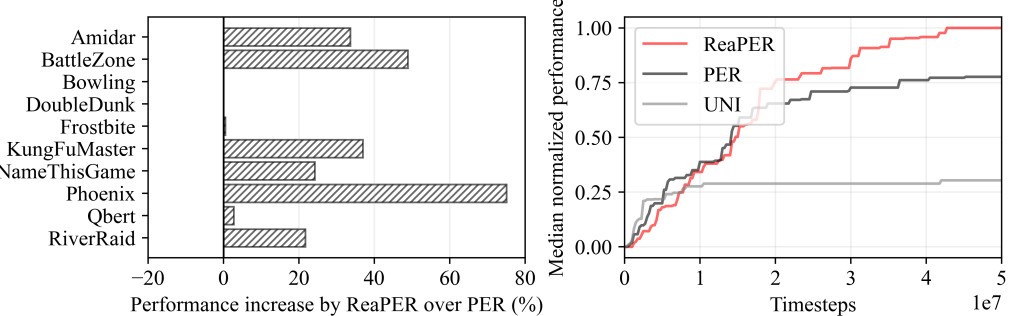

Figure 3: Left: Peak score increase of ReaPER over PER. Right: Median of the normalized cumulative maximum of scores across the Atari-10 benchmark for ReaPER, PER and uniform experience replay (UNI), following reporting standards from Schaul et al. (2015). The normalized score at timestep $t$ is calculated by dividing the difference between the current score and the random score by the difference between the maximum score in this game across all sampling strategies and the random score.

**Atari**    ReaPER consistently outperformed both uniform experience replay and PER on the ATARI-10 benchmark. Specifically, ReaPER outperformed PER and uniform experience replay in eight out of ten games, tying PER in two games. Across all games, ReaPER achieved a a $22.97\%$ higher median peak score than PER, and a $229.78\%$ higher median peak score than uniform experience replay. Under partial observability, ReaPER's median outperformance grows to $34.98\%$. These results underline ReaPER's robustness across heterogeneous game dynamics and its ability to scale to challenging, high-dimensional domains. We provide per-game curves in Appendix H (Figure 7), and extended information on results under partial observability in Appendix I.

**Discussion**    ReaPER consistently outperforms PER, indicating a substantial methodological advance. Notably, ReaPER did so with minimal hyperparameter tuning. We expect further gains through more extensive tuning of key hyperparameters, including regularization exponents $\alpha$ and $\omega$, importance sampling exponent $\beta$ and learning rate $\eta$.

A limitation of ReaPER is its reliance on terminal states, which are a pre-requisite for calculating meaningful TDE reliabilities. Further, ReaPER tracks the episodic cumulative sums of TDEs to calculate the reliability score, which causes computational overhead when TDEs are updated. Using a naive implementation, this overhead is non-negligible at $O(N)$. However, it can be reduced to

$O(n - t)$ by only re-calculating the episodic cumulative sums for transitions on their update or the update of a preceding transition within the same episode.

## 6 CONCLUSION

We introduced ReaPER, a reliability-adjusted experience replay method that mitigates the detrimental effects of unreliable targets in off-policy deep reinforcement learning. By formally linking target bias to downstream temporal difference errors, we proposed a principled reliability score that enables more efficient and stable sampling. Our theoretical analysis shows that ReaPER improves both convergence speed and variance reduction over standard PER, and our empirical results confirm its effectiveness across diverse benchmarks.

Beyond its immediate practical gains, ReaPER highlights the importance of accounting for target reliability in experience replay, particularly in deep RL settings where function approximation errors and generalization artifacts are prevalent. We believe our work opens new avenues for incorporating uncertainty and reliability estimates into replay buffers, and future research may explore adaptive reliability estimation, extensions to actor-critic methods and infinite-horizon settings, as well as integration with representation learning.

## 7 REPRODUCIBILITY STATEMENT

We provide extensive details of experimental settings and hyperparameters to reproduce our experimental results in Appendix G. Source code for all experiments is available in the supplementary materials, and will be open sourced.

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

## A   ALGORITHM

---

**Algorithm 2:** Deep Q Learning with reliability-adjusted proportional prioritization

---

**Input:** batch size $k$, learning rate $\eta$, replay period $K$, replay buffer size $N$, exponents $\alpha$, $\omega$ and $\beta$, budget $T$.

1   Initialize replay memory $\mathcal{H} = \emptyset$, $\Delta = 0$, $p_1 = 1$, episode vector $\phi = \mathbf{0}^{(N)}$, episodic count $\Phi = 1$ and maximum sum of episodic TDE $F = 1$;

2   Observe $S_1$ and choose $A_1 \sim \pi_\theta(S_1)$;

3   **for** $c \in \{1, \dots, T\}$ **do**

4      Initialize accumulated weight change $\Delta = 0$ and empty batch $\mathcal{X} = \mathbf{0}^{(k)}$;

5      Observe $S_{c+1}$, $R_c$, $d_c$;

6      Store transition $C_c = (S_c, A_c, R_c, d_c, S_{c+1})$ in $\mathcal{H}$ with $\phi_c = \Phi$ and $p_c = \max_t(p_t)$ for all $t \in \{1, \dots, N\}$;

7      **if** $c \equiv 0 \mod K$ **then**

8         **for** $m \in \{1, \dots, k\}$ **do**

9            Sample a transition $C_j$ from $\mathcal{H}$ to add to batch $\mathcal{X}$ such that $\mathbb{P}(C_t = \mathcal{X}_m) = p_t$ for all $t \in \{1, \dots, N\}$;

10            Compute importance-sampling weight $w_j = \frac{(N \cdot p_j)^{-\beta}}{\max_t w_t}$ for all $t \in \{1, \dots, N\}$;

11            Compute TDE $\delta_j = Q_{target}(S_j) - Q(S_j, A_j)$;

12            Accumulate weight-change $\Delta \leftarrow \Delta + w_j \cdot \delta_j \cdot \nabla_\theta Q(S_j, A_j)$;

13         **end**

14         Update weights $\theta \leftarrow \theta + \eta \cdot \Delta$;

15         From time to time, copy weights into target network, $\theta_{target} \leftarrow \theta$;

16         Update maximum sum of absolute TDEs, $F \leftarrow \max_{t \in \{1, \dots, N\}} \left( \sum_{i=1}^{N} \delta_i^+ \cdot \mathbb{1}_{\phi_t = \phi_i} \right)$;

17         **for** $t \in \{1, \dots, N\}$ **do**

18            Compute TDE reliabilities,

$$\mathcal{R}_t = \begin{cases} 1 - \left( \frac{\sum_{i=t+1}^{n} \delta_i^+}{\sum_{i=1}^{n} \delta_i^+} \right) & \text{for transitions of terminated episodes} \\ 1 - \left( \frac{F - \sum_{i=1}^{t} \delta_i^+}{F} \right) & \text{for transitions of ongoing episodes} \end{cases}$$

19            Update transition sampling criterion $\Psi_t \leftarrow \mathcal{R}_t^\omega \cdot \delta_t^{+\alpha}$;

20            Update transition priorities $p_t \leftarrow \frac{\Psi_t}{\sum_{i=1}^{N} \Psi_i}$;

21         **end**

22      **end**

23      **if** $d_c = 1$ **then**

24         **for** $t \in \{1, \dots, N\} \mid \phi_t = \phi_c$) **do**

25            Compute TDE reliabilities for the finished episode, $R_t = 1 - \left( \frac{\sum_{i=t+1}^{n} \delta_i^+}{\sum_{i=1}^{n} \delta_i^+} \right)$;

26         **end**

27         $\Phi \leftarrow \Phi + 1$;

28      **end**

29      Choose action $A_c \sim \pi_\theta(S_c)$;

30 **end**

---

In the following, we describe how ReaPER operates in conjunction with a Deep Q-Network (DQN). The agent begins by observing the initial state and selecting an action (Algorithm 2, Line 2).

For a fixed number of iterations, the agent interacts with the environment, observes the resulting transition from its latest action, and stores this transition in the replay buffer with maximum priority (Algorithm 2, Line 5f.).

Every $K$ steps, the agent performs a training update (Algorithm 2, Line 7). During training, it samples a batch $\mathcal{X}$ from the buffer using the current priorities $p$ as sampling weights (Algorithm 2, Line 8ff.). The agent updates the model parameters using importance-sampling-weighted TD-errors (Algorithm 2, Line 14), and uses the observed TD-errors to update the priorities $p$ for all transitions

in the batch (Algorithm 2, Line 17ff.). This involves recomputing the reliabilities based on the new TD-errors, applying a conservative estimate for transitions from ongoing episodes (Algorithm 2, Line 18). The agent then recalculates the sampling criterion $\Phi$ and updates the priorities $p$ accordingly (Algorithm 2, Line 20), concluding the training step.

Upon episode termination, the agent replaces the preliminary reliability estimate with the actual reliability (Algorithm 2, Line 25). Throughout training, it tracks episode progress to enable continuous recomputation of reliabilities (Algorithm 2, Line 27).

At each iteration, the agent selects the next action based on its current policy and state (Algorithm 2, Line 29), initiating the next cycle.

## B SUPPLEMENTARY BACKGROUND

**Target networks.** Target networks stabilize DQN by maintaining a separate copy of the Q-network whose parameters are updated more slowly. This reduces harmful feedback loops between rapidly changing estimates and improves training stability. In practice, the target network parameters are either periodically copied from the online network or updated via a soft update rule, where parameters slowly track the online network through Polyak averaging. These techniques help ensure that the bootstrap targets change smoothly over time, making optimization substantially more robust. For further information, see Mnih et al. (2015).

**DDQN.** DDQN mitigates Q-value overestimation by decoupling action selection from value evaluation: the online network selects the action, while the target network evaluates it. This leads to more accurate value estimates and often improves policy quality. In addition to reducing positive bias, DDQN can yield more stable learning dynamics, particularly in environments with noisy rewards or large action spaces, where overestimation errors tend to accumulate. For further details, see van Hasselt et al. (2015).

**Optimizers.** We use standard adaptive gradient methods. Adam (Kingma & Ba, 2014) maintains per-parameter first- and second-moment estimates to provide smooth, scaled updates. RMSProp adapts learning rates based on an exponentially weighted average of recent squared gradients, helping control step sizes in non-stationary settings. In practice, these methods reduce the sensitivity to hyperparameter choices such as the initial learning rate and improve convergence speed, especially in deep reinforcement learning where gradient magnitudes can vary significantly across parameters and time. We refer to Ruder (2017) for a comprehensive introduction.

## C LITERATURE REVIEW

In the following, we substantiate our claim that PER constitutes the most practically relevant prioritized sampling strategy within reinforcement learning to this day. We first systematically review state-of-the-art RL algorithms and the sampling strategies they are employing. We further discuss possible reasons for the limited adoption of proposed alternatives.

### C.1 PRIORITIZED EXPERIENCE REPLAY AS THE STATE-OF-THE-ART

When we refer to PER as most practically relevant sampling strategy, we do not claim that DDQN with PER, as proposed in the original PER paper Schaul et al. (2015), represents the state-of-the-art in solving RL problems overall. Rather, we claim that to this day, no transition selection algorithm within experience replay has demonstrated efficiency improvements comparable to those of PER, without incurring significant computational overhead. This claim is supported by the fact that most state-of-the-art RL algorithms use PER, while other prioritization strategy are scarcely used within state-of-the-art RL algorithms (Panahi et al., 2024).

To support this notion, we have compiled Table 1, which lists leading algorithms on the Atari benchmark since the introduction of PER, and indicates whether they use experience replay and PER.

The findings within Table 1 indicate that all state-of-the-art RL algorithms that rely on experience replay also rely on PER. The sole exception of this is Dreamer-v3 (Hafner et al., 2023), which relies

| Algorithm (Year) | Authors [Year] | Uses ER | Uses PER |
|---|---|---|---|
| Rainbow | Hessel et al. (2017) | Yes | Yes |
| Ape-X DQN | Horgan et al. (2018) | Yes | Yes |
| MuZero | Schrittwieser et al. (2019) | Yes | Yes, for the Atari benchmark |
| R2D2 | Kapturowski et al. (2019) | Yes | Yes |
| Go-Explore | Ecoffet et al. (2019) | No | No |
| NGU | Badia et al. (2020b) | Yes | Yes |
| Agent57 | Badia et al. (2020a) | Yes | Yes |
| EfficientZero | Ye et al. (2021) | Yes | Yes |
| Bigger, Better, Faster | Schwarzer et al. (2023) | Yes | Yes |
| Dreamer-v3 | Hafner et al. (2023) | Yes | No, but PER boosts performance[2] |
| SR-SPR | D'Oro et al. (2023) | Yes | Yes |
| EfficientZero-v2 | Wang et al. (2024) | Yes | Yes |

Table 1: Overview of state-of-the-art reinforcement learning algorithms, highlighting whether they utilize Experience Replay (ER) and Prioritized Experience Replay (PER).

on uniform sampling for ease of implementation, but explicitly states PER to boost performance. This provides evidence that PER remains the de-facto standard prioritized sampling strategy, and therefore represents a key point of reference for our study.

### C.2 SYSTEMATIC REVIEW OF PROPOSED ALTERNATIVES

While we discussed alternative prioritized sampling strategies to provide a comprehensive overview of related work, these methods have seen limited adoption and are not integrated into state-of-the-art reinforcement learning algorithms. We lay out potential reasons for the sparse adoption of the approaches mentioned in the paper, and thereby discuss why we do not consider them a relevant baseline for the present paper.

While we have reviewed alternative prioritized sampling strategies to provide a comprehensive overview of related work, these methods have seen limited adoption and have not been widely integrated into state-of-the-art reinforcement learning algorithms. We outline possible factors contributing to their limited uptake and explain why, in the context of this study, we do not consider them appropriate baselines.

*Ramicic & Bonarini (2017)* proposed entropy-based sampling. Their evaluation focused on a single, non-standard environment and did not include a comparison against PER. To the best of our knowledge, the authors also did not release code, which may limit the ease of direct application.

*Gao et al. (2021)* proposed reward-based sampling and evaluated their approach in two environments, FETCHREACH-V1 and PENDULUM-V0. While these experiments provide useful insights, no evaluation was presented on more complex domains such as Atari games, making direct comparison with the original PER study less straightforward. From a theoretical perspective, an emphasis on rewards could potentially bias the algorithm toward greedier behavior, which might pose challenges in more complex settings. To the best of our knowledge, code was not made publicly available, which may limit immediate applicability.

*Brittain et al. (2019)* proposed a refinement to PER by propagating priorities back through the sequence of transitions. Their evaluation compared the approach to a proportional variant of PER with a different parameterization than the $\alpha = .5, \beta = .5$ setting recommended in the original paper, which may have influenced baseline performance. To the best of our knowledge, the work was released as a preprint in 2019 but has not appeared in a peer-reviewed venue, making it more difficult to fully gauge the impact of the proposed method.

*Zha et al. (2019)* introduced Experience Replay Optimization, a dynamic prioritization approach, and evaluated it on eight continuous control environments using DDPG. Their method was compared

---

[2]While the classic Dreamer-v3 algorithm does not use PER but uniform sampling, the authors explicitly report PER to improve performance. To directly quote Hafner et al. (2023): "While prioritized replay (Schaul et al., 2015) is used by some of the expert algorithms we compare to and we found it to also improve the performance of Dreamer, we opt for uniform replay in our experiments for ease of implementation."

against PER and demonstrated improved performance, though not within the original experimental setting of the PER paper. Key hyperparameters such as $\alpha$ and $\beta$ were not reported, and, to the best of our knowledge, the authors did not release code, which may limit the reproducibility and practical applicability of their results.

*Oh et al. (2021)* introduced the Neural Experience Replay Sampler (NERS), which frames sample selection as a reinforcement learning problem by training a separate agent. While this approach is conceptually appealing, it introduces notable computational overhead, which may affect its practicality. The evaluation was conducted on Atari games for 100,000 timesteps, rather than the conventional 50,000,000, providing insights into the early stages of learning. The authors report improvements over PER in this regime; however, details on the PER configuration are not provided, and, to the best of our knowledge, the implementation has not been released, which may limit reproducibility.

## D    DETAILED FORMAL ANALYSIS

We subsequently theoretically explore the properties of ReaPER. This section extends the formal analysis in Section 3.2.

### D.1    CONVERGENCE BEHAVIOR

In the following, we provide a formal motivation for ReaPER by analyzing the influence of target bias on convergence behavior. We do so by showing that a misaligned target may degrade the value function, and then provide a decomposition of the expected error update.

**Lemma D.1** (Update misalignment due to target bias). *Let* $\mathbf{g}_t = \nabla_\theta(Q(S_t, A_t) - Q_{target}(S_t))^2$ *denote the gradient of the TDE loss and let* $\mathbf{g}_t^\star = \nabla_\theta(Q(S_t, A_t) - Q^\star(S_t, A_t))^2$ *be the ideal gradient that aligns with the true value error. Then,*

$$\langle \mathbf{g}_t, \mathbf{g}_t^\star \rangle = 2(Q(S_t, A_t) - Q^\star(S_t, A_t))^2 - 2(Q(S_t, A_t) - Q^\star(S_t, A_t))\varepsilon_t. \tag{19}$$

*Proof of Lemma D.1.* We compute the gradients explicitly. We define

$$e_t := Q(S_t, A_t) - Q^\star(S_t, A_t), \quad \varepsilon_t := Q_{\text{target}}(S_t) - Q^\star(S_t, A_t). \tag{20}$$

We now may rewrite the TDE,

$$\delta_t = Q_{\text{target}}(S_t) - Q(S_t, A_t) = (Q^\star(S_t, A_t) + \varepsilon_t) - Q(S_t, A_t) = -e_t + \varepsilon_t. \tag{21}$$

Now, the gradients are

$$\mathbf{g}_t = 2(Q(S_t, A_t) - Q_{\text{target}}(S_t))\nabla_\theta Q = 2(-\delta_t)\nabla_\theta Q(S_t, A_t), \tag{22}$$

$$\mathbf{g}_t^\star = 2(Q(S_t, A_t) - Q^\star(S_t, A_t))\nabla_\theta Q(S_t, A_t) = 2e_t \nabla_\theta Q(S_t, A_t). \tag{23}$$

Hence,

$$\langle \mathbf{g}_t, \mathbf{g}_t^\star \rangle = 4(-\delta_t)e_t\|\nabla_\theta Q(S_t, A_t)\|^2 = 4(e_t - \varepsilon_t)e_t\|\nabla_\theta Q(S_t, A_t)\|^2. \tag{24}$$

Simplifying yields

$$\langle \mathbf{g}_t, \mathbf{g}_t^\star \rangle = 4(e_t^2 - e_t\varepsilon_t)\|\nabla_\theta Q(S_t, A_t)\|^2, \tag{25}$$

which proves the result up to a constant factor of the norm.    □

This result shows that even when the TDE is large, its usefulness critically depends on the reliability of the target value. When $\varepsilon_t$ is large, the update may not improve the value function estimation. When $\varepsilon_t$ is sign-misaligned with the current true estimation error $e_t$, the update will even degrade the value function, pushing $Q(S_t, A_t)$ further away from $Q^\star(S_t, A_t)$.

Based on these considerations, we proceed to compare various sampling strategies by analyzing the expected change in the squared Q-value error caused by a single update step. The following lemma provides a decomposition of this change and builds the foundation of our main theoretical result.

**Lemma D.2** (Expected error update under sampling strategy $\mu$)**.** *Let $e_t = (Q(S_t, A_t) - Q^\star(S_t, A_t))$. Let $Q$ denote the Q-function before an update, and let $Q'$ denote the Q-function after the update. Let $\mathbb{E}_\mu[\Delta\|Q(S_t, A_t) - Q^\star(S_t, A_t)\|^2] = \mathbb{E}_\mu[\|Q'(S_t, A_t) - Q^\star(S_t, A_t)\|^2 - \|Q(S_t, A_t) - Q^\star(S_t, A_t)\|^2]$. Then,*

$$\mathbb{E}_\mu\big[\Delta\|Q(S_t, A_t) - Q^\star(S_t, A_t)\|^2\big] = 2\eta \sum_{t=1}^{n} \mu_t \mathbb{E}\big[(Q(S_t, A_t) - Q^\star(S_t, A_t))\varepsilon_t\big]$$

$$+ \eta^2 \sum_{t=1}^{n} \mu_t \mathbb{E}[\delta_t^2] - 2\eta \sum_{t=1}^{n} \mu_t \mathbb{E}[e_t^2]. \tag{26}$$

*Proof of Lemma D.2.* We analyze the Q-value update

$$Q'(S_t, A_t) = Q(S_t, A_t) + \eta\delta_t. \tag{27}$$

After the update

$$Q'(S_t, A_t) - Q^\star(S_t, A_t) = Q(S_t, A_t) + \eta\delta_t - Q^\star(S_t, A_t) = e_t + \eta\delta_t, \tag{28}$$

the squared error becomes

$$(Q'(S_t, A_t) - Q^\star(S_t, A_t))^2 = (e_t + \eta\delta_t)^2 = e_t^2 + 2\eta e_t \delta_t + \eta^2 \delta_t^2. \tag{29}$$

The expectation under sampling distribution $\mu$ is

$$\mathbb{E}[\Delta e_t] = \eta^2 \mathbb{E}[\delta_t^2] + 2\eta \mathbb{E}[e_t \delta_t]. \tag{30}$$

Note that $\delta_t = Q_{\text{target}}(S_t) - Q(S_t, A_t) = \varepsilon_t - e_t$, so

$$e_t \delta_t = e_t(\varepsilon_t - e_t) = e_t \varepsilon_t - e_t^2, \tag{31}$$

hence

$$\mathbb{E}[\Delta e_t] = \underbrace{\eta^2 \mathbb{E}[\delta_t^2]}_{(1)} - \underbrace{\mathbb{E}[e_t^2])}_{(2)} + \underbrace{2\eta(\mathbb{E}[e_t \varepsilon_t]}_{(3)}. \tag{32}$$

Summing over all transitions with $\mu_t$ gives the result. □

This decomposition highlights three components: (1) the variance of the TDE, (2) the true squared error and (3) the bias-error-interaction. The latter is key to explaining why ReaPER outperforms other sampling strategies.

A key factor in the reliability of bootstrapped targets is the extent of downstream TDEs. Intuitively, if future states still exhibit significant TDEs, the bootstrapped target for the current state is more likely to be biased. This motivates the following technical assumption.

## D.2 DISCUSSION OF ASSUMPTION 3.4

Assumption 3.4 establishes a relationship between the target bias for a given transition and the sum of TDEs for downstream transitions. This aligns with a conventional perspective in TD-learning analysis, wherein bootstrapped targets predominantly inherit bias from inaccuracies in future value estimates.

Although Assumption 3.4 appears rather limiting at first sight, it is in fact less strict than assumptions made in classical Q-learning analyses: classical Q-learning convergence proofs (see, e.g., Watkins & Dayan, 1992) rely on global exploration assumptions, ensuring that every state-action pair is visited infinitely often, and on decaying learning rates to control noise. In contrast, Assumption 3.4 takes a more local view, postulating that the target bias along an observed trajectory can be bounded by unresolved downstream TDEs. While classical assumptions ensure eventual global accuracy, our assumption focuses on bounding the bias during finite-sample learning along actual agent trajectories, which is more aligned with practical deep RL settings.

Under Assumption 3.4, the reliability score $\mathcal{R}_t$ — which measures the proportion of downstream TDE along a trajectory — provides an upper bound on the normalized target bias $\varepsilon_t$. Lemma 3.5 formalizes this relationship.

### D.3 PROOF OF LEMMA 3.5

*Proof.* From the definition of $\mathcal{R}_t$, we have

$$1 - \mathcal{R}_t = \frac{\sum_{i=t+1}^{n} \delta_i^+}{\sum_{i=1}^{n} \delta_i^+}. \tag{33}$$

Multiplying both sides by $\sum_{i=1}^{n} \delta_i^+$, we obtain

$$\sum_{i=t+1}^{n} \delta_i^+ = (1 - \mathcal{R}_t) \cdot \sum_{i=1}^{n} \delta_i^+. \tag{34}$$

Substituting this into Assumption 3.4, we find

$$|\varepsilon_t| \leq \lambda \sum_{i=t+1}^{n} \delta_i^+ = \lambda(1 - \mathcal{R}_t) \cdot \sum_{i=1}^{n} \delta_i^+, \tag{35}$$

which proves the first inequality.

Rearranging the result gives

$$\mathcal{R}_t \leq 1 - \frac{|\varepsilon_t|}{\lambda \sum_{i=1}^{n} \delta_i^+}, \tag{36}$$

completing the proof. $\square$

Moreover, it follows that

$$\mathcal{R}_t \leq 1 - \frac{|\varepsilon_t|}{\lambda \sum_{i=1}^{n} \delta_i^+}. \tag{37}$$

Lemma 3.5 provides a formal link between the reliability score $\mathcal{R}_t$ used in ReaPER and the target bias $\varepsilon_t$. Under Assumption 3.4, transitions with large downstream TDEs (i.e., large $\sum_{i=t+1}^{n} \delta_i^+$) likely suffer from higher target bias. This justifies using $\mathcal{R}_t$ to down-weight transitions with unreliable TDEs in the sampling distribution as long as downstream transitions suffer from high TDE. Consequently, ReaPER not only emphasizes transitions with high learning potential (large $\delta_t^+$) but also prioritizes those with more reliable target estimates.

### D.4 PROOF OF PROPOSITION 3.6

*Proof.* From Lemma D.2, the expected change in squared error per update, conditioned on the current Q-function, is

$$\Delta E = \eta^2 \sum_{t=1}^{n} \mu_t \mathbb{E}[\delta_t^2] - 2\eta \sum_{t=1}^{n} \mu_t \mathbb{E}[e_t^2] + 2\eta \sum_{t=1}^{n} \mu_t \mathbb{E}[e_t \varepsilon_t]. \tag{38}$$

We seek to maximize the second term (true error reduction) while minimizing the third term (bias-error-interaction). The analysis proceeds by comparing the terms under the different sampling strategies.

We now compare three sampling strategies:

*Uniform sampling* ($\mu_t = 1/n$): No prioritization occurs. Transitions with small $e_t^2$ and potentially large $e_t \varepsilon_t$ are sampled proportionally to their frequency of occurrence in the buffer. Hence, both the error reduction term $\sum_t \mu_t \mathbb{E}[e_t^2]$ and the bias term $\sum_t \mu_t \mathbb{E}[e_t \varepsilon_t]$ are solely determined by the buffer content, and sampling does not reduce error or bias.

*PER sampling* ($\mu_t \propto (\delta_t^+)$): PER prioritizes transitions with large TDE $\delta_t^+$, which correlates with larger $e_t^2$. As such, the sampling increases error reduction over buffer content,

$$\sum_{t=1}^{n} \mu_t^{\text{PER}} \mathbb{E}[e_t^2] \gg \sum_{t=1}^{n} \mu_t^{\text{Uniform}} \mathbb{E}[e_t^2], \tag{39}$$

leading to faster true error reduction compared to uniform sampling. However, PER does not account for target bias $\varepsilon_t$. Nevertheless, since PER focuses updates on transitions with large TDEs (rather than arbitrary ones), it slightly reduces the bias-error-interaction compared to uniform:

$$\sum_{t=1}^{n} \mu_t^{\text{PER}} \mathbb{E}[e_t \varepsilon_t] \ll \sum_{t=1}^{n} \mu_t^{\text{Uniform}} \mathbb{E}[e_t \varepsilon_t]. \tag{40}$$

*ReaPER sampling* ($\mu_t \propto \mathcal{R}_t \delta_t^+$): Prioritizes transitions with large TDE *and* high reliability $\mathcal{R}_t$, thus additionally considering target bias (see Lemma D.3). Thus, ReaPER achieves

$$\sum_{t=1}^{n} \mu_t^{\text{ReaPER}} \mathbb{E}[e_t^2] \gg \sum_{t=1}^{n} \mu_t^{\text{PER}} \mathbb{E}[e_t^2] \gg \sum_{t=1}^{n} \mu_t^{\text{Uniform}} \mathbb{E}[e_t^2], \tag{41}$$

for the true error term, and

$$\sum_{t=1}^{n} \mu_t^{\text{ReaPER}} \mathbb{E}[e_t \varepsilon_t] \ll \sum_{t=1}^{n} \mu_t^{\text{PER}} \mathbb{E}[e_t \varepsilon_t] \ll \sum_{t=1}^{n} \mu_t^{\text{Uniform}} \mathbb{E}[e_t \varepsilon_t], \tag{42}$$

for the bias term.

Therefore, ReaPER leads to the steepest expected decrease in squared error per update, followed by PER, followed by uniform sampling. Summing the per-step improvements over training steps, we conclude

$$\mathbb{E}_{\mu^{\text{Uniform}}}[\|Q_T - Q^\star\|^2] \geq \mathbb{E}_{\mu^{\text{PER}}}[\|Q_T - Q^\star\|^2] \geq \mathbb{E}_{\mu^{\text{ReaPER}}}[\|Q_T - Q^\star\|^2], \tag{43}$$

as claimed. □

## D.5 Discussion of Remark 3.7

While Assumption 3.4 is stated under the premise that the agent follows an optimal policy, it can be extended to fixed but suboptimal policies. We now briefly outline the modifications necessary if the agent follows a fixed but suboptimal policy. In this case, the target bias $\varepsilon_t$ cannot be bounded solely by unresolved downstream TDEs, as suboptimal actions introduce an additional, trajectory-independent bias. Formally, suppose there exists a constant $\zeta \geq 0$ such that for all transitions along observed trajectories,

$$|Q^\pi(S_t, A_t) - Q^\star(S_t, A_t)| \leq \zeta, \tag{44}$$

where $Q^\pi$ denotes the action-value function under the fixed policy $\pi$. Then, under analogous reasoning to Assumption 3.4, the target bias satisfies

$$|\varepsilon_t| \leq \lambda \sum_{i=t+1}^{n} \delta_i^+ + \zeta. \tag{45}$$

This adjusted bound propagates through the subsequent results. In particular, Lemma D.3 becomes

$$|\varepsilon_t| \leq \lambda(1 - \mathcal{R}_t) \sum_{i=1}^{n} \delta_i^+ + \zeta, \tag{46}$$

and the reliability bound adjusts accordingly. In the error decomposition of Lemma D.2 and the convergence hierarchy in Proposition D.4, the additive term $\zeta$ introduces a bias floor that does not vanish through learning. Consequently, while ReaPER still achieves improved sampling efficiency by reducing the impact of target misalignment, the achievable Q-function accuracy is ultimately lower-bounded by $\zeta$. In the limit as $\zeta \to 0$ (the policy approaches optimality), we recover the original theory.

## D.6 Proof of Proposition 3.8

*Proof.* First, we note that we can assume without loss of generality that there is a $\tau > 0$ such that the distribution $\mu^\star$ satisfies

$$\sum_{i=1}^{N} \mu_i^\star \delta_i^+ \geq \tau. \tag{47}$$

If such a $\tau > 0$ does not exist, that implies $\sum_{i=1}^N \mu_i^\star \delta_i^+$ which only holds if $\delta_i = 0$ for all $i = 1, \dots, N$ – a setting in which any sampling distribution from the buffer is optimal. Hence, equipped with (47), the optimal sampling distribution $\mu^\star$ is characterized as a solution to the following optimization problem

$$\min_{\mu \in \Delta_N} \quad \sum_{i=1}^N \mu_i \sigma_i^2 \tag{48}$$

$$\text{subject to} \quad \sum_{i=1}^N \mu_i \delta_i^+ \geq \tau. \tag{49}$$

We introduce Lagrange multipliers $\lambda \geq 0$ for the inequality constraint and $\nu$ for the probability normalization constraint. Then, the Lagrangian reads

$$\mathcal{L}(\mu, \lambda, \nu) = \sum_{i=1}^N \mu_i \sigma_i^2 + \lambda \left( \tau - \sum_{i=1}^N \mu_i \delta_i^+ \right) + \nu \left( 1 - \sum_{i=1}^N \mu_i \right) \tag{50}$$

The KKT conditions for optimality are:

$$\text{(Stationarity)} \quad \frac{\partial \mathcal{L}}{\partial \mu_i} = \sigma_i^2 - \lambda \delta_i^+ - \nu = 0 \quad \text{for all } i = 1, \dots, N \tag{51}$$

$$\text{(Primal feasibility)} \quad \sum_{i=1}^N \mu_i \delta_i^+ \geq \tau, \quad \sum_{i=1}^N \mu_i = 1, \quad \mu_i \geq 0 \tag{52}$$

$$\text{(Dual feasibility)} \quad \lambda \geq 0 \tag{53}$$

$$\text{(Complementary slackness)} \quad \lambda \left( \tau - \sum_{i=1}^N \mu_i \delta_i^+ \right) = 0 \tag{54}$$

In the following we distinguish two cases.

**Case 1:** Suppose $\sum_{i=1}^N \mu_i \delta_i^+ > \tau$. Then, complementary slackness implies $\lambda = 0$. The stationarity condition becomes

$$\sigma_i^2 - \nu = 0 \quad \Rightarrow \quad \sigma_i^2 = \nu \quad \text{for all } i = 1, \dots, N, \tag{55}$$

which is only possible if all $\sigma_i^2$ are equal. In general, this is not the case, so the constraint must be active at optimality.

**Case 2:** Then $\sum_{i=1}^N \mu_i \delta_i^+ = \tau$, and complementary slackness implies $\lambda > 0$. From the stationarity condition, we obtain

$$\sigma_i^2 - \lambda \delta_i^+ - \nu = 0 \quad \Rightarrow \quad \sigma_i^2 = \lambda \delta_i^+ + \nu \tag{56}$$

Solving for $\lambda$, we get

$$\lambda = \frac{\sigma_i^2 - \nu}{\delta_i^+} \tag{57}$$

This must hold for all $i = 1, \dots, N$, so the right-hand side must be constant across $i$, which implies

$$\frac{\sigma_i^2}{\delta_i^+} - \frac{\nu}{\delta_i^+} = \text{constant} \quad \overset{\text{(a)}}{\Rightarrow} \quad \mu_i^\star \propto \frac{\delta_i^+}{\sigma_i^2} \tag{58}$$

To justify the implication (a), note that the stationarity condition directly states $\sigma_i^2 - \nu = \lambda \delta_i^+ \Rightarrow \frac{1}{\lambda} = \frac{\delta_i^+}{\sigma_i^2 - \nu}$. Therefore, a higher value $\frac{\delta_i^+}{\sigma_i^2 - \nu}$ implies a higher gradient contribution exactly where $\mu_i$ should be large. $\qquad \square$

## E  EMPIRICAL VALIDATION OF ASSUMPTION 3.4

To empirically support Assumption 3.4, we conducted a simulation leveraging a BLINDCLIFFWALK experiment. The BLINDCLIFFWALK environment is a stylized RL setting introduced in the original

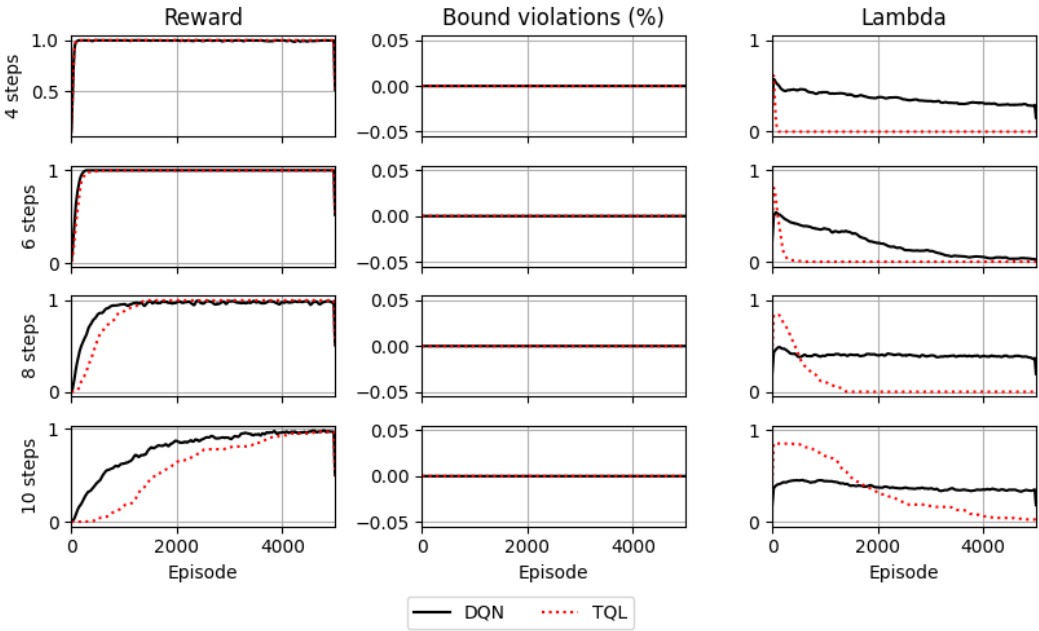

Figure 4: Rewards, bound violations and $\lambda$ in a 4-, 6-, 8-, and 10-step BLINDCLIFFWALK environment for TQL and DQN, averaged across 100 runs and smoothened across 50 consecutive data points.

PER paper (Schaul et al., 2015), which we adopt to enable comparability with prior work. In this environment, an agent has to make $n$ consecutive correct binary decisions to obtain a reward of 1. If the incorrect action in a given state is selected, the agent is reset to the initial state, incurring a reward of 0. In this controlled environment, the true Q-values can be computed exactly, enabling direct empirical evaluation of our theoretical bound. We evaluated both a Tabular Q-learning (TQL) setup and a simple DQN with a single 16-neuron hidden layer. Each algorithm was evaluated along four different cliffwalk lengths $(4, 6, 8, 10)$ for 100 iterations, with 5,000 episodes of training time.

We report average rewards as an indicator of convergence. We further report observed bound violations, defined as

$$|\varepsilon_t| > \sum_{i=t+1}^{n} \delta_i^+. \tag{59}$$

Additionally, we report the empirically observed $\lambda$, defined as

$$\lambda = \frac{|\varepsilon_t|}{\sum_{i=t+1}^{n} \delta_i^+}. \tag{60}$$

As shown in Figure 4, our results indicate that the bound established in Assumption 3.4 is never violated, neither in the TQL nor in the DQN setting. Moreover, we observe that $\lambda$ decreases as the model converges. This trend is especially pronounced in the tabular setting, yet also apparent in the DQN setting, and provides empirical support for our theoretical claim that the bound tightens as the agent's policy improves. Taken together, these findings suggest that the bound is reliable, even under complex non-tabular function-approximation dynamics.

## F    CONVERGENCE IN A STYLIZED SETTING

We consider a single episode within a stylized setting. The agent is following the optimal path, $Q_{\text{target}}(S_t) = Q(S_{t+1}, A_{t+1})$ where $A_{t+1} = \pi^\star(S_{t+1})$ for all $t \in \{1, \ldots, n-1\}$. At the end of this

| Parameter | TQL | DQN |
|---|---|---|
| Learning rate | 0.5 | 0.01 |
| Gamma | 0.99 | 0.99 |
| Exploration | Random tie break | $\epsilon = 1$ until first reward, then $\epsilon = 0$ |
| Buffer size | - | 1000 |
| Batch size | 1 | 10 |

Table 2: Hyperparameters for the TQL and DQN conditions for empirical testing of Assumption 3.4 in the BLINDCLIFFWALK environment.

episode, the agent obtains a final reward $R_n = 1$. There are no intermediary rewards. The agent aims to learn the correct Q-values for all transitions within this trial using a TQL approach (Watkins & Dayan, 1992). $Q_{\text{target}}(S_t)$ are continuously updated to be $Q(S_{t+1}, A_{t+1})$, with $Q_{\text{target}}(S_n)$ being 1. We consider a transition $C_t$ to be learned if the Q-value reaches its (for real applications mostly unknown) ground-truth Q-value, $Q(S_t, A_t) = Q^\star(S_t, A_t)$. We consider the model to have converged when all Q-values reach their ground-truth Q-value, $Q(S_t, A_t) = Q^\star(S_t, A_t)$ for $t \in \{1, \ldots, n\}$. This is the case when all TDEs within this trial have a value of 0, i.e., $\delta_t = 0$ for $t \in \{1, \ldots, n\}$.

The agent learns by repeatedly selecting $k$ transition indices. For every selected transition $C_j$, the Bellman equation is solved to update the Q-value, $Q(S_j, A_j) \leftarrow Q(S_j, A_j) + \gamma \cdot \delta_j$. For the sake of simplicity, we assume a discount factor $\gamma = 1$, a batch size $k = 1$, and a learning rate $\eta = 1$. As $\eta = 1$, learning on a transition $C_j$ implies setting the Q-value to $Q_{\text{target}}$, i.e., $Q(S_j, A_j) = Q_{\text{target}}(S_j)$. We repeat this iterative process of sampling and the respective Q-value adaptation until the model converged.

We use three different selection strategies to identify the transition index $j$, which determines the transition $C_j$ that the model trains on next, uniform sampling, Greedy Prioritized Experience Replay (PER-g), and Greedy Reliability-adjusted Prioritized Experience Replay (ReaPER-g). Uniform sampling selects transitions at random with equal probability. PER-g selects the transition with the highest TDE $\delta$. ReaPER-g selects the transition with the highest reliability-adjusted TDE $\Psi$. in both PER-g and ReaPER-g, if there is no unique maximum, ties are resolved by random choice.

We compare these selection strategies to the optimal solution, the *Oracle*. The Oracle selects the transition $C_l$ with the highest index that has a absolute TDE greater than zero, $l = \max(t \mid \delta_t \neq 0)$ for $t \in \{1, \ldots, n\}$. Given $\eta = 1$, using the Oracle, the agent will always converge within $\sum_{t=1}^{n} \mathbb{1}_{\delta_t \neq 0} \leq n$ steps and is therefore optimal. We compare the sampling strategies under varying levels of $Q_{\text{target}}$ reliability. $Q_{\text{target}}$ reliability here is determined by the extent of target Q-values for unlearned transitions without immediate reward varying from zero: Reliability is high if $Q_{\text{target}}$ values remains close to the immediate observed reward unless specifically learned otherwise. $Q_{\text{target}}$ reliability decreases with more $Q_{\text{target}}$ values deviating from zero without observation of an immediate reward and without being explicitly learned. In reality, this may happen either through Q-value initialization or - more importantly, when using Q-functions - erroneous Q-value generalization across state-action pairs.

In the present example, we simulate different levels of $Q_{\text{target}}$ reliability using different $Q$-value initializations. Specifically, we consider three $Q_{\text{target}}$ reliability conditions: High, medium and low $Q_{\text{target}}$ reliability. In all conditions, all $Q(S_t, A_t)$ for $t \in \{1, \ldots, n\}$ are first initialized to zero. Then, depending on the reliability condition, some of these initializations are overwritten with ones to induce unreliability. In the *low reliability* condition, every second Q-value is overwritten. In the *medium reliability* condition, every fourth Q-value is overwritten. In the *high reliability* condition, no value is overwritten.

We employ every selection mechanism to train until convergence for episodes of length 10 to 100 across all reliability conditions. As shown in Figure 2, uniform sampling converges the slowest across all reliability conditions. PER-g finds the optimal transition selection order when target reliability is high. However, PER-g's convergence speed quickly diminishes as $Q_{value}$ reliability decreases. ReaPER-g on the other hand actively accounts for changes in $Q_{value}$ reliability. By doing so, it consistently finds the ideal solution regardless of $Q_{value}$ reliability.

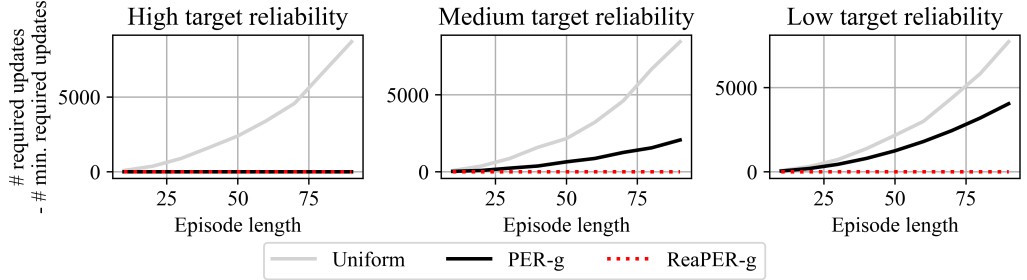

Figure 5: Performance comparison in a stylized setting for three sampling methods; uniform sampling, PER-g, and ReaPER-g. Performance is quantified as the number of updates until convergence minus the minimally required number of updates given by an Oracle. Each sampling method is evaluated using different episode lengths (between 10 and 100) and different levels of $Q_\text{target}$ reliability (high, medium and low).

## G    EXPERIMENTAL SETTINGS

All training parameters for all environments were set to according to pre-existing recommendations from previous research (Mnih et al., 2015; Schaul et al., 2015; van Hasselt et al., 2015; Raffin, 2020). The full experimental settings are subsequently described in detail to enable full reproducibility.

**Experience replay parameters**    Following the suggestion for proportional PER for DDQN in Schaul et al. (2015), $\alpha$ was set to 0.6 for PER. As we expect increased need for regularization due to the reliability-driven priority scale-down, we expected values smaller values for $\alpha$ and $\omega$ in ReaPER. We performed minimal hyperparameter tuning to find a suitable configuration. We did so by training a single game, QBERT, on three configurations for $\alpha$ and $\omega$, (1) $\alpha = 0.2$, $\omega = 0.4$, (2) $\alpha = 0.3$, $\omega = 0.3$ and (3) $\alpha = 0.4$, $\omega = 0.2$. For the runs presented in the paper, the best-performing variant ($\alpha = 0.4$, $\omega = 0.2$) was used. For both PER and ReaPER, $\beta$ linearly increased with training time, $\beta \leftarrow (0.4 \rightarrow 1.0)$ as proposed in Schaul et al. (2015). A buffer size of $10^6$ was used.

**Atari preprocessing**    As in Mnih et al. (2015), Atari frames were slightly modified before being processed by the network. Preprocessing was performed using StableBaselines3's *AtariWrapper* (Antonin Raffin et al., 2024). All image inputs were rescaled to an 84x84 grayscale image. After resetting the environment, episodes were started with a randomized number of *NoOp*-frames (up to 30) without any operation by the agent, effectively randomizing the initial state and consequently preventing the agent from learning a single optimal path through the game. Four consecutive frames were stacked to a single observation to provide insight into the direction of movement. Additionally, a termination signal is sent when a life is lost. All these preprocessing steps can be considered standard practice for ATARI games (e.g., Mnih et al. (2013; 2015); Schaul et al. (2015)).

**Training specifications**    Hyperparameters for learning to play CARTPOLE, ACROBOT and LU-NARLANDER were set to RL Baselines3 Zoo recommendations (Raffin, 2020). Hyperparameters for learning to play *Atari* games are set based on previous research by Schaul et al. (2015), Mnih et al. (2015) and van Hasselt et al. (2015).

**Network architecture**    For CARTPOLE, ACROBOT and LUNARLANDER, the network architecture was equivalent to *StableBaselines3*' default architecture (Antonin Raffin et al., 2024): A two-layered fully-connected net with 64 nodes per layer was used. For image observations as in the ATARI-10 <benchmark, the input was preprocessed to size $84 \times 84 \times 4$. It was then passed through three convolutional layers and two subsequent fully connected layers. The network architecture was equal to the network architecture used in Mnih et al. (2015). Rectified Linear Units (Agarap, 2018) were used as the activation function.

---

[3]In the uniform experience replay condition, the learning rate was increased to 2.5e-4, as recommended in seminal work (Mnih et al., 2015; Schaul et al., 2015; van Hasselt et al., 2015).

| Parameter | CARTPOLE | ACROBOT | LUNARLANDER | ATARI |
|---|---|---|---|---|
| Learning rate | 2.3e-3 | 6.3e-4 | 6.3e-4 | 625e-5[3] |
| Budget in timesteps | 5e4 | 1e5 | 1e5 | 5e7 (2e8 frames) |
| Buffer size | 1e5 | 5e4 | 5e4 | 1e6 |
| Timestep to start learning | 1e3 | 1e3 | 1e3 | 5e4 |
| Target network update interval | 10 | 250 | 250 | 3e4 |
| Batch size | 64 | 128 | 128 | 32 |
| Steps per model update | 256 | 4 | 4 | 4 |
| Number of gradient steps | 128 | 4 | 4 | 1 |
| Exploration fraction | 0.16 | 0.12 | 0.12 | 0.02 |
| Initial exploration rate | 1 | 1 | 1 | 1 |
| Final exploration rate | 0.04 | 0.1 | 0.1 | 0.01 |
| Evaluation exploration fraction | 0.001 | 0.001 | 0.001 | 0.001 |
| Number of evaluations | 100 | 100 | 100 | 200 |
| Trajectories per evaluation | 5 | 5 | 5 | 1 |
| Gamma | 0.99 | 0.99 | 0.99 | 0.99 |
| Max. gradient norm | 10 | 10 | 10 | $\infty$ |
| Reward threshold | 475 | -100 | 200 | $\infty$ |
| Optimizer | Adam | Adam | Adam | RMSprop |

Table 3: Comprehensive documentation of hyperparameters used within the CARTPOLE, ACROBOT, LUNARLANDER and ATARI environments.

**Evaluation** For the environments CARTPOLE, ACROBOT and LUNARLANDER, 100 evaluations were evenly spaced throughout the training procedure. Each agent evaluation consisted of five full trajectories in the environment, going from initial to terminal state. The evaluation score of a single agent evaluation is the average total score across those five evaluation trajectories. Training was stopped when the agent reached a predefined reward threshold defined in the Gymnasium package (Towers et al., 2024). For ATARI environments, as in Schaul et al. (2015), 200 evaluations consisting of a single trajectory were evenly spaced throughout the training procedure. No reward threshold was set.

**Score normalization**: For ATARI games, scores were normalized to allow for comparability between games. Let $\Xi_{raw}$ denote a single evaluation score that is to be normalized. Let $\Xi_{random}$ denote the score achieved by a randomly initialized policy in this game. Let $\Xi_{max}$ denote the highest score achieved in this game across either condition, ReaPER or PER. The normalized score $\Xi_{norm}$ is then calculated via

$$\Xi_{norm} = \frac{\Xi_{raw} - \Xi_{random}}{\Xi_{high} - \Xi_{random}}. \tag{61}$$

**Percentage improvement**:

For the environments CARTPOLE, ACROBOT and LUNARLANDER, we compute percentage improvements in median timesteps-to-convergence $\mathcal{T}$ until meeting a specified reward threshold of ReaPER upon another condition $\mathcal{C}$, as stated within Section 5, as

$$\text{Improvement over } \mathcal{C} = \frac{\mathcal{T}_{\mathcal{C}} - \mathcal{T}_{\text{ReaPER}}}{\mathcal{T}_{\mathcal{C}}} \times 100. \tag{62}$$

For the ATARI environments, we compute percentage improvements in peak score $\mathcal{P}$ of ReaPER upon another condition $\mathcal{C}$, as stated within Section 5, as

$$\text{Per-game improvement over } \mathcal{C} = \frac{\mathcal{P}_{\text{ReaPER}} - \mathcal{P}_{\mathcal{C}}}{\mathcal{P}_{\mathcal{C}}} \times 100. \tag{63}$$

The overall improvement across the ATARI-10 benchmark is computed as the median of the improvements observed on each individual game.

## H ATARI-10 RESULTS

Figure 6 displays the median of normalized scores across games gathered throughout 200 evaluation periods, which were evenly spaced-out throughout the 50 million training timesteps.

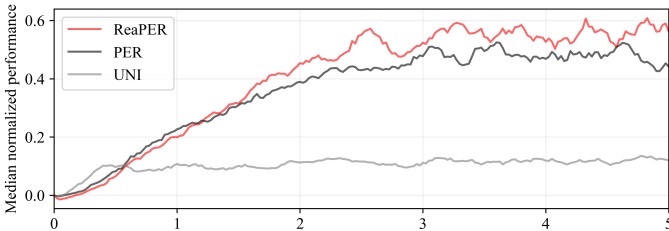

Figure 6: Median of normalized scores across the ATARI-10 benchmark for ReaPER, PER and uniform experience replay (UNI) with a moving average smoothed over 10 points.

Figure 7 displays the cumulative maximum scores per game gathered across 200 evaluation periods.

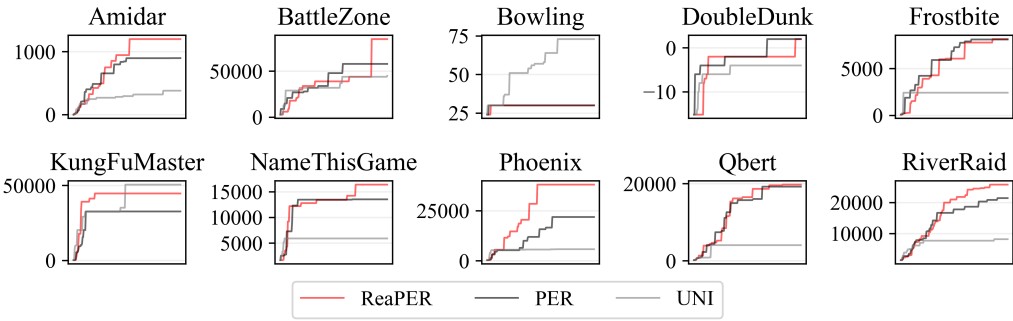

Figure 7: Cumulative maximum evaluation score per game from the ATARI-10 benchmark across the training period for ReaPER, PER and uniform experience replay (UNI).

## I PARTIAL OBSERVABILITY

In Atari environments, a single-frame observation does not convey object velocities or movement directions, making the vanilla problem setting a partially observable MDP (Hausknecht & Stone, 2017). Standard practice mitigates this by stacking four consecutive frames as the agent's observation to ease the learning process (Mnih et al., 2015). To perform a deliberate study of performance under partial observability, we intentionally restrict the agent to a single-frame observation.

Under this constraint, ReaPER outperforms PER in 8 of 10 games, achieving a median relative improvement of 34.97%, substantially exceeding the 22.97% gain observed in the fully observable setting. This result indicates that the benefits of reliability adjustment are further amplified under partial observability. Aggregated results are shown in Figure 8. Per-game curves are shown in Figure 9.

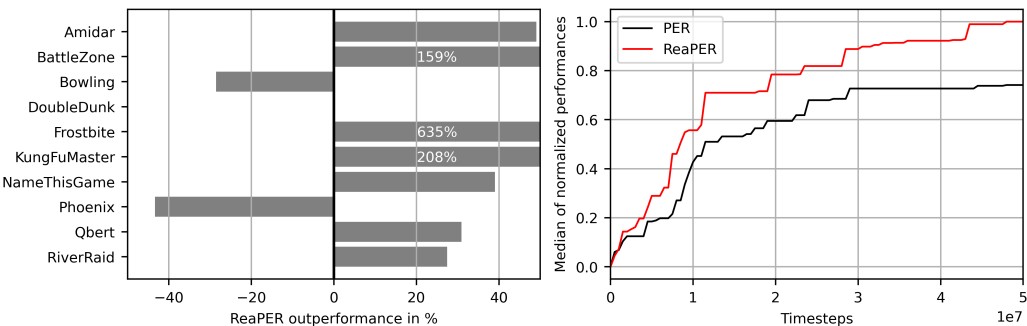

Figure 8: Left: Peak score increase of ReaPER over PER under partial observability, induced by single-frame observations. Right: Median of the normalized cumulative maximum of scores across the Atari-10 benchmark for ReaPER and PER under partial observability. The normalized score at timestep $t$ is calculated by dividing the difference between the current score and the random score by the difference between the maximum score in this game across all sampling strategies and the random score.

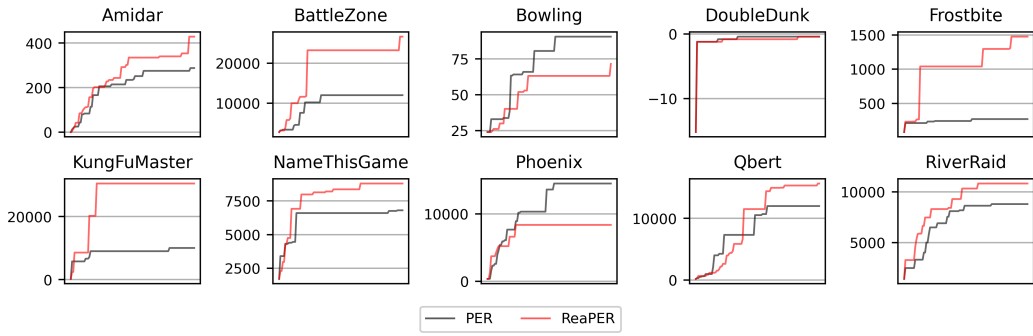

Figure 9: Cumulative maximum evaluation score per game from the ATARI-10 benchmark across the training period for ReaPER and PER under partial observability, induced by single-frame observations.

## J    HARDWARE SPECIFICATION

The ATARI experiments were conducted on a workstation equipped with an AMD Ryzen 9 7950X CPU (32 cores at 4.5 GHz), 128 GB of RAM, and an NVIDIA RTX 4090 GPU with 24 GB of memory (driver version 12.3). All other numerical experiments were performed on a 2024 MacBook Air with an Apple M3 processor.

