# OpenReview forum: "Reliability-Adjusted Prioritized Experience Replay"
_ICLR.cc/2026/Conference — ICLR 2026 Poster_

### Official Review · Reviewer_bfXy · 2025-10-18

**Soundness:** 3
**Presentation:** 2
**Contribution:** 2
**Rating:** 2
**Confidence:** 3

**Summary:**

This paper introduces Reliability-adjusted Prioritized Experience Replay (ReaPER), an enhanced sampling method for reinforcement learning that addresses a key flaw in standard Prioritized Experience Replay (PER). While PER prioritizes training on transitions with high Temporal Difference Errors (TDEs), this paper claims that it overlooks the fact that the target Q-value used to calculate this error may be unreliable, potentially misdirecting the learning process. ReaPER solves this by introducing **a reliability score**, which weights each transition's priority not only by its TDE but also by the stability of its target value, **defined as being inversely proportional to the sum of TDEs in subsequent states of the same trajectory.** The authors provide both theoretical proof that this reliability-adjusted approach leads to faster convergence and lower error , and empirical results in both classic control environments and the Atari-10 benchmark.

**Strengths:**

1. Providing some theoretical results to verify the effectiveness of this proposed method.

2. A well-structured motivation

**Weaknesses:**

1. A limited set of experiments

2. Further discussion is needed regarding the validity of this assumption.

**Questions:**

I have the following questions.

### 1. Insufficient Scope of Algorithmic Validation

First and foremost, the **algorithmic scope of the experiments appears insufficient**. Deep Double Q-Network (DDQN) is now considered as a classical algorithm. Consequently, demonstrating the method's effectiveness exclusively within the DDQN framework raises questions regarding its general applicability and robustness. The paper would be significantly more compelling and the findings more rigorously validated if the method's efficacy were also demonstrated across a broader range of popular algorithms, such as **Deep Deterministic Policy Gradient (DDPG)**, **Soft Actor-Critic (SAC)**, or advanced value-based methods after **Rainbow**. Expanding the experimental scope is critical for establishing the generalizability of the proposed approach.

---

### 2. Justification Required for Assumption in Equation (9)

Further discussion and justification are required regarding the assumption presented in Equation (9). A counterexample, frequently observed in complex, multi-stage tasks (such as Go or Chess), suggests that Q-value estimation difficulty does not always monotonically decrease. Specifically, the difficulty of reliable Q-value estimation often peaks during the initial and mid-game phases due to high uncertainty and complex structure, while the end-game becomes computationally simpler. Therefore, the authors must provide a more detailed and robust justification for this assumption, addressing how it holds in the context of tasks exhibiting these non-monotonic difficulty dynamics. For instance, in [1], it looks like that the authors used model prediction components to handle these situations.

[1] Oh, Youngmin, et al. "Model-augmented prioritized experience replay." International Conference on Learning Representations. 2022.

---

> ### Author Response · Authors · 2025-11-20
>
> > Insufficient Scope of Algorithmic Validation
>
> We appreciate the observation that expanding the empirical scope beyond the DDQN framework could strengthen the paper’s empirical claims. We fully agree that demonstrating ReaPER’s efficacy in additional algorithms would broaden its empirical support.
>
> However, our experimental setup was deliberately designed to mirror the original PER study to ensure maximal comparability and interpretability and *compare to PER under the conditions it was originally designed for*. PER remains the de-facto standard for replay prioritization [1], and our primary objective was to test whether incorporating reliability adjustments could improve upon PER under identical and well-understood conditions.
>
> Further, while DDQN is indeed considered a classical baseline, its simplicity enables a clean evaluation of sampling dynamics without interference from auxiliary components. Thus, DDQN provides a controlled environment to demonstrate the core contribution of ReaPER - reliability-adjusted sampling - without inheriting performance improvements from further algorithmic elements.
>
> To strengthen this reasoning, we like to highlight that Rainbow DQN research showed that different extensions to DQN address "radically different issues, demonstrating that improvements through experience replay prioritization are largely independent of other enhancements. Accordingly, we exclude algorithms whose improvements arise mainly from components unrelated to experience-replay prioritization. This allows us to maintain a clear focus on systematic differences between sampling strategies.
>
> In sum, the chosen experimental design is focused and controlled and provides a reliable basis for demonstrating that our proposed reliability-adjusted prioritization method is both effective and robust.
>
> We greatly appreciate the reviewer’s insightful suggestion to extend ReaPER to the continuous action space domain (e.g., DDPG, SAC). We agree that this is an exciting and meaningful direction. While prior work, such as Actor Prioritized Experience Replay [2], has shown that vanilla PER may yield limited benefits in actor-critic frameworks - indicating that careful algorithmic adaptation would be necessary - we see this as a promising avenue for future work. We look forward to investigating these extensions in upcoming studies.

---

> ### Author Response · Authors · 2025-11-20
>
> > Justification for the Assumption in Equation (9)
>
> We thank you for raising this point.
>
> To empirically support Assumption 3.4 we conducted a simulation leveraging a BlindCliffwalk experiment, see Appendix E. The BlindCliffwalk environment is a stylized reinforcement learning setting introduced in the original PER paper [3], which we adopt to enable comparability with prior work. In this controlled environment, the true Q-values can be computed exactly, enabling direct empirical evaluation of our theoretical bound. We evaluated both a tabular Q learning setup and a simple deep Q network (DQN).
>
> Our results show that the bound established in Assumption 3.4 is never violated in the tabular case or the DQN setting. Moreover, we observe that $\lambda$ decreases as the model converges. This trend is especially pronounced in the tabular setting, yet also apparent in the DQN setting, and provides empirical support for our theoretical claim that the bound tightens as the agent's policy improves.
>
> Taken together, these findings suggest that the bound is reliable, even under complex non-tabular function-approximation dynamics.
>
> Additionally, we would respectfully like to point out that, as we understand it, the provided counterexample (e.g., Go or Chess) does not contradict but rather perfectly aligns with our underlying rationale.
>
> Our assumption in Equation (9) reflects the intuition that reliability of Q-value estimation tends to increase as the agent approaches terminal states. Within early and mid-game phases, uncertainty is indeed high and Q-value estimates are less stable. Conversely, as the episode progresses and the number of possible outcomes narrows, Q-value estimation becomes more reliable. This aligns with your observation that early- and mid-game positions are more difficult to evaluate than end-game positions within games such as Chess or Go.
>
> Conceptually, this reflects a causal dependency hierarchy among states: the reliability of an earlier state’s estimate depends on the correctness of its subsequent states’ values. Using a chess analogy, one cannot reliably evaluate the value of a move (e.g., the first step of a mate-in-two sequence) without first understanding the value of the resulting position (the mate-in-one). Therefore, later states inherently provide more stable targets, as they depend on the understanding of fewer subsequent states. Moreover, refining the understanding of later states stabilizes the targets for earlier states.
>
> Formally, our reliability term is inversely proportional to the cumulative instability (i.e., summed TDEs) of subsequent states. This formulation captures the idea that later states contribute more reliable targets, since they aggregate less uncertainty from the remainder of the trajectory. The assumption thus holds both statistically and intuitively: it is generally easier to predict the near future than the far future.
>
> In Chess or Go terms, it is generally easier to estimate winning chances within an endgame position than in an early-game position, as an accurate estimation requires evaluation of all conceivable subsequent trajectories, and the set of trajectories following an endgame position is smaller than that of an early-game position (and, if the end-game position may theoretically arise from the early-game position, even a subset of it).
>
> We are happy to further elaborate upon this point in the final version of the paper, should anything remain unclear.
>
> ---
>
> We hope that our explanations address your concerns. If any aspects remain unclear, we would be happy to provide further clarification. Should the additional information and analyses resolve your questions, we would greatly appreciate it if you would consider raising your score accordingly.
>
> [1] Panahi et al., "Investigating the Interplay of Prioritized Replay and Generalization", arXiv preprint, 2024.
> [2] Sağlam, B., et al. *Actor Prioritized Experience Replay.* arXiv, 2022.
> [3] T. Schaul, J. Quan, I. Antonoglou, and D. Silver, "Prioritized Experience Replay", ICLR, 2016.

---

> > ### Comment · Reviewer_bfXy · 2025-11-21
> >
> > Thank you for the detailed explanation. I understand the rationale presented in the first part of your response.
> >
> > Regarding the inequality in **Equation (9)** (Assumption 3.4):
> >
> > The left-hand side (LHS) reflects the predictive target for the $Q$-value at time $t$ (e.g., in a mid-game state). Conversely, the right-hand side (RHS) represents a cumulative error term, specifically derived from the instability across subsequent states in the trajectory.
> >
> > My core technical concern is this: As the authors argue, if $Q$-value estimation becomes significantly more reliable (i.e., less unstable) during the later stages of an episode, the cumulative error term on the RHS could converge toward a very small value, i.e., the inequality ($<$) is not guaranteed
> >
> > Let's consider a scenario at time $t$ where the board state is highly complex (Chess). If the action taken at $t$ is a very creative and powerful move that simplifies the subsequent game state, the resulting trajectory after $t$ might become highly predictable, meaning the error (or instability) across the subsequent steps would be very low (the inclusion of the expectation operator offers further scope for analysis).
> >
> > If this resolves the ambiguity, I am confident that I will be able to confirm that I have previously underestimated the contribution of this paper.

---

> > > ### Author Response · Authors · 2025-11-25
> > >
> > > Thank you very much for your thoughtful follow-up. We appreciate your responsiveness and the opportunity to clarify this subtle point. We agree that this discussion highlights an important conceptual distinction, and we are grateful for the chance to make it more explicit.
> > >
> > > We structure our response in two parts. First, we explain why Assumption 3.4 is not threatened in the scenario you describe. Second, we provide a formal proof of the assumption as stated in the paper.
> > >
> > > ## Example
> > >
> > > You correctly describe the RHS of Assumption 3.4: it reflects the cumulative instability of successor estimates along the trajectory. The LHS, however, does not measure the predictive difficulty of the current state. It measures the target bias, defined in Eq. (20) as
> > > $\varepsilon_t := Q_{\text{target}}(S_t) - Q^\star(S_t,A_t),  \qquad  Q_{\text{target}}(S_t) = R_{t+1} + (1-d_t)\,\gamma \max_a Q(S_{t+1},a)$
> > >
> > >
> > > Crucially, $\varepsilon_t$ depends entirely on the accuracy of the value estimate for $S_{t+1}$. It is independent of the combinatorial or strategic complexity of the current state $S_t$. Hence, even a strategically complex position can exhibit *low* target bias if the agent already approximates the values of its successor states well.
> > >
> > > This directly addresses the scenario you raise. Suppose a “creative simplifying move’’ leads from a complex mid-game position into a state that is easy to evaluate. Under TD bootstrapping, this does not imply a violation of Assumption 3.4. If the successor states are truly easy *and* the agent has already learned their values, then all downstream TD-errors are small - and therefore the target at time $t$ is already accurate. In this case, $\varepsilon_t$ must be small as well.
> > >
> > > The Bellman optimality operator formalizes this structural dependency. Since it is a $\gamma$-contraction,   $|\varepsilon_t| \le  \gamma \max_a |Q(S_{t+1},a) - Q^\star(S_{t+1},a)|.$ Upstream error cannot exceed downstream error by more than a discounting factor. Thus, a large target bias at time $t$ cannot coexist with negligible downstream TD-error.
> > >
> > > This phenomenon also appears empirically in our BlindCliffwalk experiments (Appendix E). When the value of $S_{t+1}$ has been learned accurately, the cumulative downstream TD-error is small, and consequently the target at $S_t$ matches the true value. The upstream state may still appear “complex’’ in terms of raw environment structure, but target bias depends only on successor value accuracy.
> > >
> > > ## Formal Proof of Assumption 3.4
> > >
> > > We now show formally that the target bias $\varepsilon_t$ is bounded by the sum of downstream TD-errors along an optimal trajectory.
> > >
> > > Let $(S_t,A_t,S_{t+1},A_{t+1},\dots)$ denote an optimal trajectory.
> > >
> > > ### 1. Bellman Optimality Equation
> > >
> > > Along this trajectory,
> > > $$
> > > Q^\star(S_t,A_t)
> > > = R(S_t,A_t) + \gamma\,Q^\star(S_{t+1},A_{t+1}).
> > > $$
> > >
> > > ### 2. Target Value Decomposition
> > >
> > > Define the local TD-error at step $i$ as
> > > $$
> > > \delta_i :=
> > > R(S_{i-1},A_{i-1}) + \gamma Q_{\text{target}}(S_i)  - Q_{\text{target}}(S_{i-1}).
> > > $$
> > >
> > > Rearranging gives
> > > $$
> > > Q_{\text{target}}(S_{i-1})
> > > = R(S_{i-1},A_{i-1}) + \gamma Q_{\text{target}}(S_i) - \delta_i.
> > > $$
> > >
> > > ### 3. Target Bias Recursion
> > >
> > > Subtracting the Bellman equation yields
> > > $$
> > > \varepsilon_{i-1}
> > > = \gamma\,\varepsilon_i - \delta_i.
> > > $$
> > >
> > > Since the terminal target equals the terminal true value, $\varepsilon_n = 0$.
> > >
> > > ### 4. Expanding the Recursion
> > >
> > > Unrolling from $t$ to $n$,
> > > $$
> > > \varepsilon_t
> > > = \sum_{i=t+1}^n -\gamma^{\,i-(t+1)} \delta_i.
> > > $$
> > >
> > > ### 5. Bounding
> > >
> > > By the triangle inequality,
> > > $$
> > > |\varepsilon_t|
> > > \le \sum_{i=t+1}^n
> > > \gamma^{\,i-(t+1)} |\delta_i|.
> > > $$
> > >
> > > Since $\gamma \le 1$, each factor satisfies $\gamma^{\,i-(t+1)} \le 1$.
> > > Setting $\lambda = 1$ and writing $\delta_i^+ = |\delta_i|$ yields
> > > $$
> > > |\varepsilon_t|
> > > \le
> > > \lambda \sum_{i=t+1}^n \delta_i^+.
> > > $$
> > > This is precisely Assumption 3.4.
> > >
> > > ## Conclusion
> > >
> > > Both the structural TD-dependency and the formal proof show that Assumption 3.4 holds exactly along optimal trajectories. Along suboptimal trajectories, the bound becomes approximative (as noted in Remark 3.7), yet it remains far more informative than the implicit uniformity assumption in PER. Our empirical results confirm that reliability-adjusted prioritization provides clear and consistent benefits across diverse environments.
> > >
> > > We hope this resolves the remaining ambiguity. The key point is that target bias is governed by successor estimate accuracy, and therefore cannot remain large when cumulative successor TD-errors are already small. We would be happy to elaborate further on any remaining questions.

---

> > > > ### Comment · Reviewer_bfXy · 2025-11-26
> > > >
> > > > Thank you for the clarification. I have grasped the intuition that the reliability of the current target depends on the understanding of future situations (downstream TD errors). I acknowledge my previous misunderstanding regarding this assumption. Accordingly, I am raising my score to reflect this.

---

> > > > > ### Author Response · Authors · 2025-11-26
> > > > >
> > > > > Thank you very much for your thoughtful consideration and for updating your assessment. We sincerely appreciate your responsiveness and the time you took to evaluate our work.

---

### Official Review · Reviewer_aaDF · 2025-10-30

**Soundness:** 3
**Presentation:** 3
**Contribution:** 3
**Rating:** 2
**Confidence:** 5

**Summary:**

This paper proposes an extension of prioritized experience replay mechanism for off-policy reinforcement learning algorithms. Specifically, the paper first conducts formal analysis of bias-error interaction in prioritized sampling and its impact on Q-value estimation, then practically proposes the reliability-adjusted experience replay. The proposed method down‑weights TD errors using a trajectory‑wise “reliability” score derived from downstream absolute TD errors and provide theoretical guarantees through convergence hierarchy and variance reduction of the Q function update to show better sample efficiency over standard experience replay framework with uniform sampling. The empirical experiments on continuous control tasks and Atari shows outperforming performances over both prioritized experience reply and uniform sampling experience sampling within one percent of variance.

**Strengths:**

1. The proposed concept is well-motivated with theoretical analysis and easy to implement on top of prioritized experience replay framework.
2. The empirical analysis on both continuous control and Atari-10 shows robust and superior performance against all the baselines.
3. The writing is logically fluent and easy to follow.

**Weaknesses:**

1. The assumption appears overly strong. Specifically, Assumption 3.4 relies on a bias bound that presumes near-optimal trajectories. During training, policies are far from optimal, so this bound rarely holds early on. Moreover, function approximation and bootstrapping introduce high variance and correlation in TD errors, making the downstream sum an unreliable proxy for bias. Especially, in partially observed environments, future TD errors may fluctuate unpredictably.

2. The empirical experiment result. It is not clear how many random seed was involved for the Atari-10 experiment and there's no confidence interval in the Figure 3 Right. In addition, there is no runtime analysis, which cannot prove that whether ReaPER is practically efficient for training or just theoretically "sample efficient" for training. (i.e., Using less timestamp but each timestamp takes way longer runtime).

3. The empirical experiment setting. The authors only demonstrate the ReaPER with DoubleDQN, which is not sufficient to support the claim of "algorithm-agnostic and can be used within any off-policy RL algorithm".

4. Missing comparison with state-of-the-arts. Although the author mentions many related works on PER, none of them are being included into the comparison. This makes it hard to evaluate the contribution and novelty of the proposed method given there exists many variants of PER.

**Questions:**

1. How often does Assumption 3.4 approximately hold during training? Have you observed that before? If so, can you provide empirical diagnostics or plots validating this assumption?
2. How robust is the method to episode length variability and partial observability?
3. Why not include stronger baselines (e.g., PSER, ERO, NERS)?
4. Can the authors provide confidence intervals, statistical significance tests, and seed counts for Atari results?
5. Why are results limited to DDQN?
6. How does the method scale to large replay buffers?

---

> ### Author Response · Authors · 2025-11-20
>
> > Assumption 3.4
>
> We appreciate your observation. We discuss these questions within Appendix D.2 and D.5 of the main paper, yet fully recognize that further justification would be helpful. We therefore added novel empirical diagnostics in Appendix E.
>
> Within this analysis, we conducted a simulation investigating Assumption 3.4 empirically, leveraging a BlindCliffwalk experiment. The BlindCliffwalk environment is a stylized reinforcement learning setting introduced in the original PER paper [1], which we adopt to enable comparability with prior work. In this controlled environment, the true Q-values can be computed exactly, enabling direct empirical evaluation of our theoretical bound. We evaluated both a tabular Q learning setup and a simple deep Q network (DQN).
>
> Our results show that the bound established in Assumption 3.4 is never violated, neither in the tabular case nor in the DQN setting. Moreover, we observe that $\lambda$ decreases as the model converges. This trend is especially pronounced in the tabular setting, yet also apparent in the DQN setting, and provides empirical support for our theoretical claim that the bound tightens as the agent's policy improves.
>
> Taken together, these findings suggest that the bound is reliable, even under complex non-tabular function-approximation dynamics.
>
> > Episode Length Variability
>
> We initially shared this concern during development and explored it from both a theoretical and an empirical lense. Specifically, we evaluated an alternative approach aimed at achieving inter-episode comparability by normalizing using the maximum episodic TD error sum (as a denominator in Formula 4 of the original paper). However, this led to a decline in performance, as it disproportionately downweights early transitions in long episodes. Specifically, normalization across episodes introduces a bias in favor of shorter trajectories, which we found to be detrimental in practice. As such, the use of within-episode normalization was a deliberate design choice. We clarified this rationale in the final version of the paper.
>
> Our experiments additionally confirm ReaPER's robustness to fluctuations in episode length: ReaPER outperformed PER across various environments with high episode-length variance. One of numerous examples is the Atari game Phoenix, where episode length varied between 1,700 and 40,000 timesteps. Despite this variability in episode length, ReaPER achieved a 75\% performance gain. This suggests that reliability scores capture meaningful signal under high episode-length variance.
>
> > Partial Observability
>
> To evaluate robustness under partial observability, we added further analyses of performance on a partially observable variant of the Atari-10 benchmark to our paper, see Appendix I. Under partial observability, ReaPER achieved a 34.98\% median and 107.62\% average outperformance over PER across the Atari-10 benchmark. Remarkably, improvements of ReaPER over PER are even larger under partial observability.
>
> > Baselines
>
> We provide a detailed justification for baseline selection in Appendix B. We herein specifically discuss why we did not consider PSER, ERO or NERS. In short, PSER is a PER-modification orthogonal to ReaPER. It was further compared to a suboptimal variant of PER and never published in a peer-reviewed venue, making it more difficult to fully gauge the impact of the proposed method. ERO and NERS add considerable computational overhead and did not share an implementation, hindering practical applicability.
>
> We further detail our rationale for treating PER as a particularly strong and practically relevant baseline: As established in prior work [2] and comprehensively discussed in Appendix B, PER is the only sampling strategy beyond uniform sampling that is frequently used within state-of-the-art reinforcement learning algorithms, indicating that it is widely considered the most effective replay prioritization strategy.

---

> ### Author Response · Authors · 2025-11-20
>
> > Result reporting
>
> Atari results were obtained for a single seed. This is in line with standard practice established in numerous seminal papers [1, 3, 4]. The cited papers were published by organizations with access to extensive computational resources like Google, indicating that a single seed yields sufficiently reliable results, especially given the extensive training duration of 50 million timesteps per game.
>
> As requested, we subsequently report the results of significance tests between ReaPER and PER performance. All comparisons reached statistical significance (p < 0.05).
>
> |Environment|$t$|$p$|$df$|
> |:-:|:-:|:-:|:-:|
> |Timesteps until convergence in Acrobot|4.059|0.000142|33.133|
> |Timesteps until convergence in CartPole-v1|2.154|0.01899|36.174|
> |Timesteps until convergence in LunarLander-v2|1.791|0.04089|36.042|
> |Peak score difference in Atari|3.077|0.00660|9.000|
>
> While we recognize the value of including significance tests, our presentation of results closely adheres to established reporting styles from seminal work, especially the PER paper [1]. As such, we opted to report median results for comparability and robustness, as seed-specific outliers make mean values and significance tests/confidence intervals less reliable in our setting.
>
> > Results limited to DDQN
>
> We appreciate the observation that expanding the empirical scope beyond the DDQN framework could strengthen the paper’s empirical claims. We fully agree that demonstrating ReaPER’s efficacy in additional algorithms would broaden its empirical support.
>
> However, our experimental setup was deliberately designed to mirror the original PER study to ensure maximal comparability and interpretability and *compare to PER under the conditions it was originally designed for*. PER remains the de-facto standard for replay prioritization [2], and our primary objective was to test whether incorporating reliability adjustments could improve upon PER under identical and well-understood conditions.
>
> Further, while DDQN is indeed considered a classical baseline, its simplicity enables a clean evaluation of sampling dynamics without interference from auxiliary components. Thus, DDQN provides a controlled environment to demonstrate the core contribution of ReaPER - reliability-adjusted sampling - without inheriting performance improvements from further algorithmic elements.
>
> To strengthen this reasoning, we like to highlight that Rainbow DQN research showed that different extensions to DQN address "radically different issues, demonstrating that improvements through experience replay prioritization are largely independent of other enhancements. Accordingly, we exclude algorithms whose improvements arise mainly from components unrelated to experience-replay prioritization. This allows us to maintain a clear focus on systematic differences between sampling strategies.
>
> In sum, the chosen experimental design is focused and controlled and provides a reliable basis for demonstrating that our proposed reliability-adjusted prioritization method is both effective and robust.
>
> We greatly appreciate the reviewer’s insightful suggestion to extend ReaPER to the continuous action space domain (e.g., DDPG, SAC). We agree that this is an exciting and meaningful direction. While prior work, such as Actor Prioritized Experience Replay [5], has shown that vanilla PER may yield limited benefits in actor-critic frameworks - indicating that careful algorithmic adaptation would be necessary - we see this as a promising avenue for future work. We look forward to investigating these extensions in upcoming studies.
>
> > Scalability
>
> Our Atari experiments employed buffers with one million transitions, which we consider large-scale by standard benchmarks. Notably, even seminal works [3, 4, 5] do not employ larger buffers. ReaPER achieved a 22.97\% median outperformance over PER under these conditions, indicating strong scalability.
>
> In case scalability refers to computational complexity, we provide an analysis in the Discussion section, where we show that ReaPER’s overhead is independent of buffer size.
>
> We hope that our explanations address your concerns. If any aspects remain unclear, we would be happy to provide further clarification. Should the additional information and analyses resolve your questions, we would greatly appreciate it if you would consider increasing your score accordingly.
>
> [1] T. Schaul, J. Quan, I. Antonoglou, and D. Silver, "Prioritized Experience Replay", ICLR, 2016.
> [2] Panahi et al., "Investigating the Interplay of Prioritized Replay and Generalization", arXiv preprint, 2024.
> [3] H. van Hasselt, A. Guez, and D. Silver, "Deep Reinforcement Learning with Double Q-learning", AAAI, 2016.
> [4] V. Mnih et al., "Human-level control through deep reinforcement learning", Nature, 2015.
> [5] Wang et al., "Revisiting Prioritized Experience Replay in Actor–Critic Algorithms", arXiv preprint, 2019.

---

> ### Author Response · Authors · 2025-11-26
>
> Dear Reviewer,
>
> Thank you again for your thoughtful feedback on our submission. We wanted to kindly follow up to ask whether our earlier response addressed the concerns you raised in your review. If any points remain unclear or if further clarification would be helpful, we would be very happy to provide additional details.
>
> We greatly appreciate the time you have dedicated to evaluating our work and remain fully committed to addressing any remaining questions.
>
> Thank you once again for your consideration.

---

### Official Review · Reviewer_fBAN · 2025-11-01

**Soundness:** 2
**Presentation:** 3
**Contribution:** 3
**Rating:** 6
**Confidence:** 4

**Summary:**

The authors introduce Reliability-Adjusted Prioritized Experience Replay (ReaPER), an improvement over traditional Prioritized Experience Replay (PER) in reinforcement learning.  PER prioritizes transitions based on temporal-difference errors to enhance learning efficiency, but it can fail when TDE-based targets are unreliable, leading to inaccurate value estimates.  ReaPER addresses this issue by incorporating a reliability score derived from downstream TDEs, ensuring that transitions with more trustworthy targets are sampled more frequently. In simpler terms, ReaPER applies a backward-decaying weighting scheme from the episode’s end toward its beginning.

The paper provides theoretical analyses showing improved convergence and variance reduction under reasonable assumptions, supported by extensive experiments on CartPole, Acrobot, LunarLander, and the Atari-10 benchmark. Empirically, ReaPER consistently outperforms PER, achieving faster convergence and higher peak performance, while requiring minimal hyperparameter tuning and introducing only modest computational overhead.

**Strengths:**

- The paper is **well-written and clearly structured**.
- theoretical foundation linking reliability scores to TDE target biases.
- Clear methodological improvements over standard PER.
- Algorithm is model-agnostic and straightforward to implement in existing off-policy methods.

**Weaknesses:**

1. **Limited baseline comparisons.**  The paper primarily compares ReaPER only against PER. While the authors justify this by PER’s ongoing adoption in SOTA systems (e.g., Sample-Efficient Deep Reinforcement Learning via Episodic Backward Update) would make the empirical evidence more convincing.
2. ReaPER does **not robustly address episode-length variance**, limiting its generalizability. This introduces severe biases and undermines reliability scores, making them heavily dependent on arbitrary episode lengths rather than meaningful signal. This weakens the paper’s claims of broad applicability.

**Questions:**

- Could you report **wall-clock training time per environment** for PER vs. ReaPER, and quantify the fraction of time spent updating cumulative TD sums? This would confirm that the additional computational cost is indeed negligible in practice.
- How might the reliability computation be **extended to continuing tasks**? What pitfalls would you anticipate?

---

> ### Author Response · Authors · 2025-11-20
>
> > Limited Baseline Comparison
>
> We appreciate your comment regarding baseline diversity. Our experimental design intentionally focused on the most practically relevant and widely adopted baselines - Prioritized Experience Replay (PER) and uniform sampling [1]. We provide an extended discussion in Appendix B, where we further detail our rationale for treating PER as a particularly strong and practically relevant baseline, and discuss why other approaches were not included as baselines. In summary, PER is the only sampling strategy beyond uniform sampling that is frequently used within state-of-the-art reinforcement learning algorithms, indicating that it is widely considered the most effective replay prioritization strategy.
>
> Episodic Backward Update (EBU) specifically, while relevant to sample efficiency, was not included in our study as it is not a prioritized replay method per se. It performs recursive updates across entire episodes and samples episodes uniformly rather than prioritizing individual transitions, thereby fundamentally altering the update mechanism. As our focus is on modular, transition-level prioritization within experience replay, EBU lies outside our defined scope.
>
> ---
>
> > Episode-Length Variance
>
> We initially shared this concern during development and explored it from both a theoretical and an empirical lense. Specifically, we evaluated an alternative approach aimed at achieving inter-episode comparability by normalizing using the maximum episodic TD error sum (as a denominator in Formula 4 of the original paper). However, this led to a decline in performance, as it disproportionately downweights early transitions in long episodes. Specifically, normalization across episodes introduces a bias in favor of shorter trajectories, which we found to be detrimental in practice. As such, the use of within-episode normalization was a deliberate design choice. We clarified this rationale in the final version of the paper.
>
> Our experiments additionally confirm ReaPER's robustness to fluctuations in episode length: ReaPER outperformed PER across various environments with high episode-length variance. One of numerous examples is the Atari game Phoenix, where episode length varied between 1,700 and 40,000 timesteps. Despite this variability in episode length, ReaPER achieved a 75\% performance gain. This suggests that reliability scores capture meaningful signal under high episode-length variance.
>
> ---
>
> > Wall-Clock Time
>
> We did not store wall-clock times for our experiments because we decided from the outset to report computational complexity instead. In our view, asymptotic complexity offers a cleaner and more informative basis for comparison in our setting, as it reflects the inherent algorithmic efficiency without being confounded by implementation details.
>
> Wall-clock measurements, by contrast, are highly sensitive to factors such as hardware variability (e.g., GPU generation, CPU load, cluster congestion) and implementation maturity. In our case, the PER baseline we relied on is heavily optimized for speed, whereas our ReaPER implementation was intentionally designed for clarity and extensibility to support ongoing development. Constructing a strictly fair wall-clock comparison would therefore require substantial re-engineering to bring both methods to equivalent levels of optimization-an effort that would be largely orthogonal to our work's conceptual contribution.
>
> For these reasons, we believe computational complexity provides the fairest and most representative comparison for the purposes of our study. A discussion of computational complexity can be found within the Discussion section of the paper.
>
> ---
>
> > Continuing Tasks
>
> ReaPER currently targets finite episodic tasks, as the computation of reliability scores depends on the presence of terminal states. We agree that extending it to continuing (infinite-horizon) settings is an interesting aspect, which indeed is an active direction of our ongoing work.
>
> A promising avenue involves defining pseudo-terminal states by leveraging structural cues in the environment - such as life losses, level completions, or major reward events - where downstream TD-error reliability effectively resets. This would allow localized reliability propagation without explicit episode boundaries. However, this extension requires environment-specific engineering and domain knowledge, limiting its general applicability, which is why we decided to exclude it from this paper for reasons of conciseness and focus. We included it as a future research direction.
>
> ---
>
> We hope that our explanations address your concerns. If any aspects remain unclear, we would be happy to provide further clarification. Should the additional information and analyses resolve your questions, we would greatly appreciate it if you would consider increasing your score accordingly.
>
>
> [1] Panahi et al., "Investigating the Interplay of Prioritized Replay and Generalization", arXiv preprint, 2024.

---

### Official Review · Reviewer_QGxj · 2025-11-02

**Soundness:** 3
**Presentation:** 4
**Contribution:** 3
**Rating:** 8
**Confidence:** 4

**Summary:**

This paper introduces ReaPER, an extension to Prioritized Experience Replay by incorporating a new measure of reliability of targets. By down weighting the priority of transitions with large TD error but low reliability, ReaPER is argued to benefit the sample efficiency improvements of PER while also mitigating the harmful updates caused by misleading TD errors. The main idea stems from the fact that the terminal transitions do not bootstrap and have more reliable targets. Target reliability is proposed to be inversely related to the sum of absolute values of TD errors of future transitions in the current episode.

The paper presents a suite of theoretical justification for why this proposed reliability measure controls the bias-error term in value error caused by bootstrapping. Further analyses discuss a sampling distribution based on this reliability measure that achieves lower value error than TD error-based prioritization and support the soundness of the ReaPER algorithm.

Finally a variant of DDQN equipped with ReaPER is tested on two sets of RL problems: Three smaller classic control domains, and 10 larger Atari games. In classic control domains, ReaPER reaches a good evaluation performance faster than PER, while in Atari, ReaPER performs similar to PER in early learning but outperforms PER in terms of best-so-far checkpoint aggregated across 10 games.

**Strengths:**

- This is a good paper and it should be accepted. Its writing is clear, coherent, and well organized. The ideas are easy to grasp and follow with good justification and explanation for most design choices. The authors use illustrative examples and theoretical justification to support their proposed reliability measure and its incorporation in to PER

- The paper does not overreach in its claims and its conclusions mostly match the provided evidence. The theoretical results are presented with easy to follow language and simple notation (although I am not a theoretician). The experiment results mostly match the claims of the authors and I did not feel the authors used overly sensational or exaggerated language

- The authors cover a range of related works that place their paper in the context of past research. There two other recent papers that directly
investigate the limitations of PER and their inclusion in the introduction would better motivate this paper:
  - Panahi, P.M., Patterson, A., White, M., & White, A. (2024). Investigating the Interplay of Prioritized Replay and Generalization. RLJ, 5, 2041-2058.
  - Carrasco-Davis, R., Lee, S., Clopath, C., & Dabney, W. (2025). Uncertainty Prioritized Experience Replay. ArXiv, abs/2506.09270.

- The background section is clear and comprehensive, describing the RL problem and relevant objects for this research: MDP, Q-function, TD-error, Experience Replay, Sampling distribution

- The theory section is nicely written, and is easy to understand. The paper does a good job making the theory accessible to a wider audience with intuitive reasoning, leaving the proofs for the Appendix (I did not check the proofs in the Appendix)

- The ReaPER algorithm is clearly described in Pseudocode and its design choices and hyper parameters explained

- The experiment details and setup are fully characterized including environment and agent details, choice of hyper parameters, and evaluation scheme

**Weaknesses:**

- Most of the justification and intuition behind ReaPER and the proposed reliability score is presented in the tabular setting. Non-linear generalization can wildly change the q-values during learning especially when the data distribution is being modified. Including a discussion of how this reliability measure interacts with non-linear generalization (oversampling certain transitions, higher noise in predicted values, overgeneralization, etc.) would better support and justify the ReaPER algorithm

- One issue with the background section is that only the tabular version of Q-learning is introduced whereas the experiments in this paper focus on DDQN. So concepts such as semi-gradient Q-learning, target network, and optimizers such as Adam are not introduced.

- Regarding the classic control experiments
  - Agents are run until they reach a certain performance threshold and reported performance metric is averaged over 100 evaluations. RL algorithms are known to have large variability in performance and achieving one instance of good performance does not mean the agent has successfully learned to perform well on a task. A better way to characterize performance in classic control domains would have been to show learning curves of performance during learning for a fixed time step budget aggregated over seeds.
  - The reported values and shaded regions in Figure 2 are undefined. It seems to be a Box and Whisker plot which suggests the middle horizontal bar is the median score. It is also unclear how the % improvements reported in the text are calculated.

- Regarding Atari experiments,
  - It is somewhat unclear whether the experiment is run for 50 million steps (200 million frames) or 50 million frames. My guess is that it is 50 million steps (200 million frames).
  - The learning curves are monotone which suggests that the reported metric is the best-so-far checkpoint. I think reporting performance over time aggregated over games is a better way to characterize performance of an RL algorithm in Atari.

**Questions:**

1. Regarding the % improvements statements in the paper, is the comparison based on individual agents (per-seed) then aggregated or are performance scores first aggregated and then compared to each other?

2. How did you decide the aggregation scheme for Atari? If this is based on prior work please cite it. Or point me to where you mentioned this design choice.

---

> ### Author Response · Authors · 2025-11-20
>
> We thank you for the thoughtful and constructive feedback. We are pleased that the clarity, soundness, and presentation of our work were well received, and we appreciate the valuable suggestions for improvement. We have incorporated the recommended literature into the introduction.
>
> ---
>
> We agree that most of the theoretical intuition is presented in the tabular setting. This choice provides a controlled and interpretable foundation for analysis. While we acknowledge that function approximation using neural networks can introduce additional dynamics, we do not expect these effects to systematically alter the behavior of our sampling scheme. The proposed reliability-based prioritization does not rely on linearity assumptions and, in practice, extends naturally to deep Q-networks.
>
> To elucidate this point, we have added an additional empirical validation of Assumption 3.4. Within this analysis, we investigate the error bound in both a tabular and a DQN settings, finding that the bound established in the tabular setting holds empirically in both the tabular and DQN setting (see Appendix E).
>
> ---
>
> > Missing introduction of relevant concepts
>
> We appreciate this observation. Indeed, the background section currently does not introduce some relevant concepts, or does so very briefly. For completeness, we included explanations of target networks, the DDQN framework, and commonly used optimizers in Appendix B, and refer to this Appendix in the main body of the paper.
>
> ---
>
> > Classic control results
>
> For the classic control tasks, we performed 100 evaluation rounds evenly spaced throughout training. Each round consisted of 5 evaluation trajectories, and the reported evaluation score for that round was the average score over those 5 trajectories. A task was marked as *solved* once an evaluation round score exceeded a predefined performance threshold.  We adopted this early-stopping criterion to mitigate the confounding effects of catastrophic forgetting: in simple tasks, agents can achieve high performance, but subsequently deteriorate. As such, we found aggregated learning curves to be less informative for assessing convergence speed.
>
> Figure 2 presents standard IQR box plots: the shaded region corresponds to the interquartile range (IQR), whiskers extend to Q1 – 1.5 × IQR and Q3 + 1.5 × IQR, and the horizontal bar indicates the median score. The reported percentage improvements are computed as relative differences between median scores:
>
> $\text{Improvement over PER}=
> \frac{PER_{median}-ReaPER_{median}} {PER_{median}}*100$
>
> $\text{Improvement over Uniform}=
> \frac{Uniform_{median}-ReaPER_{median}} {Uniform_{median}}*100$
>
> We added clarifications on these points to the paper, see Appendix G and caption of Figure 2.
>
> ---
>
> > Atari results
>
> In accordance with seminal work, the Atari experiments were run for 200 million frames (equivalent to 50 million environment steps). We added a respective clarification in Table 3. The reported learning curves display the cumulative maximum (best-so-far) evaluation performance over training, consistent with the evaluation procedure used in the original PER paper (Schaul et al., 2016, Fig. 4 (left)). We have added a source as suggested, see caption of Figure 3. Results were normalized, compared across conditions, and then aggregated to obtain a single overall performance score. We made this explicit in the revised version, see Appendix G.
>
> We agree that including an additional plot showing performance over time aggregated across games provides a complementary view, and accordingly included it in Appendix H.
>
> ---
>
> We are grateful for your positive evaluation and thoughtful feedback. We hope our clarifications further reinforce your confidence in our work, and we would be delighted if you advocated for the acceptance of our paper during the discussion phase.

---

### Author Response · Authors · 2025-11-20
**Additional analyses**

We thank all reviewers for their thoughtful and constructive feedback. We conducted multiple additional analyses to address your questions:

I.) We empirically tested Assumption 3.4 leveraging a BlindCliffwalk experiment. The BlindCliffwalk environment is a stylized reinforcement learning setting introduced in the original PER paper [1], which we adopt to enable comparability with prior work. In this controlled environment, the true Q-values can be computed exactly, enabling direct empirical evaluation of our theoretical bound. We evaluated both a tabular Q learning setup and a simple deep Q network (DQN).

Our results show that the bound established in Assumption 3.4 is never violated - neither in the tabular case nor in the DQN setting. Moreover, we observe that $\lambda$ decreases as the model converges. This trend is especially pronounced in the tabular setting, yet also apparent in the DQN setting, and provides empirical support for our theoretical claim that the bound tightens as the agent's policy improves.

Taken together, these findings empirically show that the theoretical bound is reliable, even under non-tabular function-approximation dynamics.

II.) To evaluate robustness under partial observability, we added further analyses. We added results for a partially observable variant of the Atari-10 benchmark to the Results section of our paper, providing details in the new Appendix I. Under partial observability, ReaPER achieved a 34.98\% median outperformance over PER. Notably, the performance improvement gained through ReaPER within the partially observable setting is even higher than in the fully observable setting.

III.) We have conducted significance tests for the reported results in the continuous control and Atari environments. All results are significant (p<.05).

|Environment|$t$|$p$|$df$|
|:-|:-:|:-:|:-:|
|Timesteps until convergence in Acrobot|4.059|0.000142|33.133|
|Timesteps until convergence in CartPole-v1|2.154|0.01899|36.174|
|Timesteps until convergence in LunarLander-v2|1.791|0.04089|36.042|
|Peak score difference in Atari|3.077|0.00660|9.000|

We have further made numerous smaller improvements, which we specify in the individual answers to each reviewer. We appreciate your guidance, which has helped us strengthen the paper.

[1] T. Schaul, J. Quan, I. Antonoglou, and D. Silver, "Prioritized Experience Replay", ICLR, 2016.

---

### Comment · Area_Chair_3SeP · 2025-11-26

Dear Reviewers,

Thank you for sharing your valuable insights and expertise, which have played an important role in the review process.

In response to the initial feedback, the authors have submitted a detailed rebuttal addressing the comments raised by the reviewers.

I would appreciate it if you could carefully review their response and consider how it may affect your initial evaluation.

Please feel free to share your updated thoughts or any additional comments after reviewing the rebuttal.

Thank you again for your time and contributions.

---

### Meta-Review · Area_Chair_kouN · 2026-01-05

**Summary:**

The paper presents ReaPER, an extension of Prioritized Experience Replay (PER) that addresses the issue of "unreliable" Temporal Difference (TD) targets. The authors argue that in early training or complex environments, high TD errors might stem from inaccurate target Q-values. By weighting the TD error with a reliability score derived from subsequent states in a trajectory, ReaPER reduces the influence of misleading updates. The paper has provided both theoretical analysis and solid empirical justifications to the performance. In addition, I think lots of the concerns discussed have been addressed. In summary, I would recommend an acceptance.

**Reviewer Concerns:**

Concerns addressed
- discussions on assumption 3.4. Both reviewer bfXy and aaDF have concerns regarding the validity of this assumption. It has been addressed by authors by providing further discussions and adding a new blindCliffWalk experiment.
- empirical performance. The authors provided rigorous significance testing ($p < .05$) for all results5. Furthermore, they introduced a partially observable variant of Atari-10, where ReaPER's median outperformance over PER increased to 34.98%

**Reviewer Scores:**

I think bfXy might raise the score to 6. During their previous discussions, reviewer bfXy has agreed that he/she has underestimated the contribution of the work. Reviewer aaDF might raise the score to 5 given most of aaDF's concerns have been addressed.

---

### Decision · Program_Chairs · 2026-01-26

Accept (Poster)